# A hierarchical, retinotopic proto-organization of the primate visual system at birth

Michael J Arcaro*, Margaret S Livingstone

Department of Neurobiology, Harvard Medical School, Boston, United States

**Abstract** The adult primate visual system comprises a series of hierarchically organized areas. Each cortical area contains a topographic map of visual space, with different areas extracting different kinds of information from the retinal input. Here we asked to what extent the newborn visual system resembles the adult organization. We find that hierarchical, topographic organization is present at birth and therefore constitutes a proto-organization for the entire primate visual system. Even within inferior temporal cortex, this proto-organization was already present, prior to the emergence of category selectivity (e.g., faces or scenes). We propose that this topographic organization provides the scaffolding for the subsequent development of visual cortex that commences at the onset of visual experience

## Introduction

The adult primate visual system comprises a series of interconnected, topographically-organized areas arranged in two, dorsal and ventral, distributed hierarchies (*Felleman and Van Essen, 1991*; *Ungerleider and Mishkin, 1982*). This global organization, as well as the anatomical location and functional properties of individual areas throughout cortex, are similar across individuals, suggesting an early common program for the development of the visual system. Here we ask what the initial organization is and to what extent does it resemble the adult hierarchical organization. Using a combination of correlation and task-evoked analyses of fMRI data, we show that the hierarchical, topographic organization of visual cortex is present at birth and therefore constitutes a proto-organization for the entire primate visual system. In newborns, patterns of correlated activity across occipital, temporal, and parietal cortices revealed the arealization of visual cortex and the specialization of these areas into dorsal and ventral visual pathways. Correlated activity between areas reflected the topographic organization of visual space (retinotopy). Further, the retinotopic organization within high-level visual areas emerged prior to other functional selectivity. Specifically, we identified eccentricity representations in inferotemporal cortex (IT) months before IT showed the adult-like clustered organization for face selectivity. Tuning for spatial frequency and low-level shape features such as curvature were also present prior to the detection of category domain clusters in IT. These results show that areal differentiation, hierarchical organization, low-level shape selectivity, and retinotopic organization of the entire visual system is present at birth, and likely serve as the scaffold for subsequent activity-dependent modifications throughout visual cortex.

## Results

### Functional specificity in the neonate visual system

We evaluated the functional organization of the visual system in neonatal macaques using fMRI. During the first few weeks after birth, visually evoked responses were present only in the LGN and

***For correspondence:**
Michael_Arcaro@hms.harvard.edu

**Competing interests:** The authors declare that no competing interests exist.

weakly in peripheral V1 (*Figure 1a*). In contrast to visually evoked responses, correlated activity between visual areas was significant even in these first few weeks, and clearly distinguished visual areas from the rest of cortex. For example, the mean signal of V1 was strongly correlated with occipital, temporal, and posterior parietal cortices during non-visual-stimulation rest periods (*Figure 1b*). We used retinotopic mapping in the same monkeys when they were more than 1.5 years old to map out the entire visual system (*Figure 1—figure supplement 1*). These maps were registered to the newborn data and confirmed that the extent of the V1 correlations in newborns covered the entire cortical visual system. The correspondence of the correlation maps to extrastriate visual areas was further verified at about 1 month after birth when visually-evoked responses could be reliably measured throughout these extrastriate areas (*Figure 1c*). Though the lack of early visually-evoked fMRI activity is in apparent contrast with previous electrophysiological findings that demonstrated visual responsiveness in infants (*Rust et al., 2002*; *Zhang et al., 2005*; *Endo et al., 2000*), this may reflect differences between physiological and metabolic measurements as a similar lack of early activity has been observed using deoxyglucose (*Distler et al., 1996*). As such, the apparent dissociation between the absence of visually-evoked activations and the presence of V1-correlated activity (*Figure 1a and b*) may reflect population-level immaturity of the visual pathway (*Distler et al., 1996*; *Bachevalier et al., 1991*) or a low-arousal state in these neonatal monkeys. Regardless of the reason for the lack of visually-evoked activity, the patterns of V1 correlations indicate that signal correlation approaches can reveal the functional organization of the newborn visual system long before any visually-evoked fMRI mapping is possible.

We found evidence for still further regionalization of the neonatal brain using data-driven independent component analysis (ICA), similar to previous studies that had identified functionally related

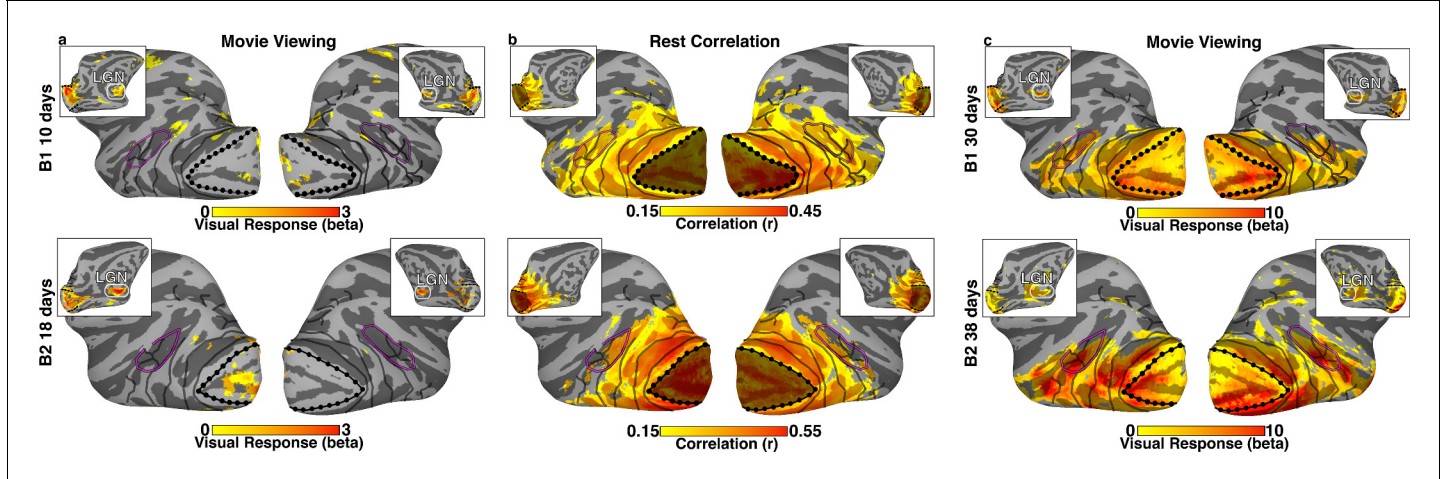

**Figure 1.** Visually–evoked activity and correlated signal in neonatal monkeys B1 and B2. (**a**) Visually evoked-activity from 30 s movie clips of faces, scenes, and scrambled images in monkeys B1 and B2 at 10 and 18 days postnatal, respectively, was largely absent on the lateral surface of visual cortex (t(782) > 1.96; p<0.05, uncorrected). As seen in the inset images of the medial surface, peripheral V1 and LGN were the only visual structures showing significant visually-evoked activity. (**b**) Within-hemisphere correlation maps on the lateral surface of the brain (inset medial surface), from a seed of the mean signal in V1 during a rest period (r > 0.15; t(1628) > 6.12 and t(1394) > 5.66 for B1 and B2, respectively; p<0.0001, FDR-corrected). Significant correlations were apparent across much of the visual system, including occipital, temporal, and parietal cortices. Dotted black lines indicate the border between V1 and V2. Shading indicates the seed area for the mean V1 signal used for calculating within-hemisphere correlations. Solid black lines mark what we could later (>1.5 years) identify as borders between retinotopic areas (*Figure 1—figure supplement 1*). Purple line corresponds to what we could later identify as the peripheral eccentricity ridge of the MT cluster (comprising areas MT, MST, FST, V4t). (**c**) Visually evoked-activity in response to 30 s movie clips of faces, scenes, and scrambled images in monkeys B1 and B2, at 30 and 38 days postnatal respectively, was present throughout the thalamus, V1, and extrastriate visual cortex (t(782) > 3.33; p<0.05, FDR corrected).

The following figure supplements are available for figure 1:

**Figure supplement 1.** Retinotopic mapping.

**Figure supplement 2.** Data-driven spatial ICA maps reveal regional organization in monkeys B1 and B2, at 10 and 18 days postnatal respectively.

areas in adult monkeys (*Hadj-Bouziane et al., 2014*; *Hutchison et al., 2011*; *Moeller et al., 2009*). Similar to these findings in adults, several visual cortical ICs were identified in B1 and B2, at 10 and 18 days old respectively, comprising foveal or peripheral occipital cortex, dorsal extrastriate, posterior parietal, middle temporal (MT), and IT cortex (*Figure 1—figure supplement 2*). Spatial maps for each IC were bilateral and symmetric, encompassing comparable regions between hemispheres. These data further indicate that functionally specific regional differentiation, both of the entire visual system and even within the visual system, was already present in newborn macaques. Also, as in adults, there were ICs in these neonatal data within somatomotor, auditory, frontal, and parietal cortices, as well as within the thalamus, demonstrating regionally-specific signals throughout sensory and association regions in the newborn brain.

## Arealization of neonate visual cortex

Even individual visual areas across occipital, temporal, and parietal cortices could be distinguished at birth using seed-based correlation analysis. We evaluated temporal correlations across hemispheres, which rules out spurious correlations between regions in close anatomical proximity due to spatial spread of the fMRI signal (*Smirnakis et al., 2007*). Previous anatomical studies in primates have shown callosal connections between the same visual areas in opposite hemispheres, that is, homotopic, in adult (*Abel et al., 2000*; *Kennedy et al., 1986*) and infant (*Felleman and Van Essen, 1991*; *Kennedy et al., 1986*; *Underleider and Mishkin, 1982*; *Maunsell and van Essen, 1983*; *Van Essen et al., 1982*; *Kaas, 1995*; *Dehay et al., 1986*; *Baldwin et al., 2012*) primates, and these callosal connections are correlated with areal and hierarchical organization (*Kennedy et al., 1986*; *Maunsell and van Essen, 1983*; *Van Essen et al., 1982*; *Kaas, 1995*; *Schmidt, 2013*). Consistent with these anatomical studies, previous fMRI studies in adult humans and monkeys have shown homotopic correlations (*Moeller et al., 2009*; *Biswal et al., 1995*; *Vincent et al., 2007*; *Butt et al.,*

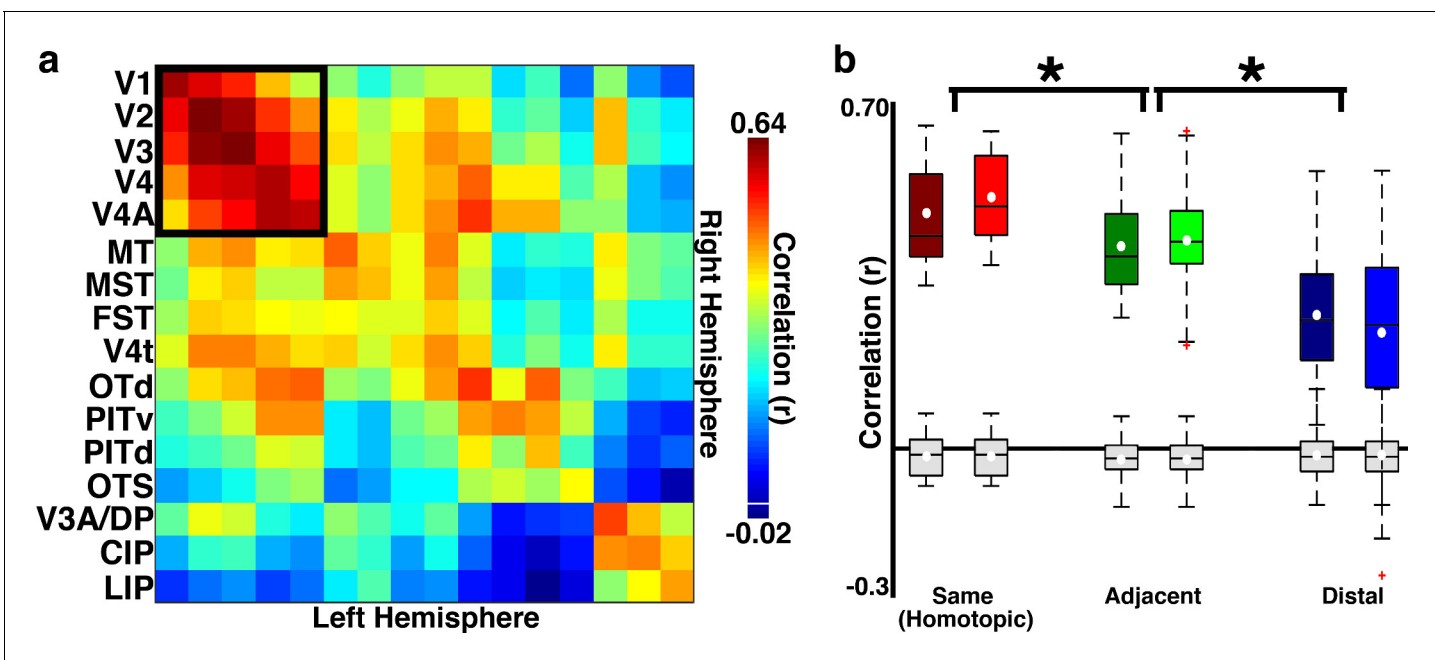

**Figure 2.** Across-hemisphere correlations reveal arealization in neonates 10 and 18 days old. (a) Across-hemisphere neonatal correlation matrix for what we could later identify as 16 visual cortical areas from retinotopic mapping (*Figure 1—figure supplement 1*). Black square illustrates the areas included in the dorsal vs. ventral analysis presented in *Figure 6*. Colorbar range scaled to min and max correlation values. White line in colorbar indicates 0. (b) Box plots for three area groups: same (homotopic) contralateral cortical areas, adjacent (to homotopic) contralateral areas, and distal (nonhomotopic and nonadjacent) contralateral areas in B1 (dark boxes) and B2 (light boxes). Mean (white circle), median (black horizontal line), interquartiles (whiskers), and outliers (red cross). There was a clear effect of area grouping (2-way ANOVA; main effect of area group: $F_{(2,506)}=92.57$, $p<0.0001$; no main effect of monkey and no interaction between monkey and area group, ps>0.1). Comparable effects were found for data averaged across hemispheres. Grey box plots show what would be expected from any biases in our fMRI acquisition or analyses (Materials and methods: Simulation of correlation due to instrumentation and analyses).

2015), which are mediated by callosal connections (*Johnston et al., 2008*; *O'Reilly et al., 2013*). Therefore, we reasoned that homotopic correlations could serve as an unbiased method for identifying regions that were already functionally distinct in neonate monkeys (i.e., areas that show strong homotopic correlations already must have functionally differentiated themselves from the surrounding cortex). Cortical areas determined by retinotopic mapping in the same monkeys at >1.5 years of age were used as seeds in the correlation analysis (*Figure 1—figure supplement 1*). Across all 16 cortical areas tested, mean signals between homotopic cortical areas were significantly correlated in each monkey (ts(15)>18.07; ps<0.0001). To compare homotopic correlations with non-homotopic correlations, across-hemisphere correlations were grouped as a function of area pair (e.g., the correlation between right hemisphere V1 and left hemisphere MT was grouped with the correlation between left hemisphere V1 and right hemisphere MT), and then grouped into homotopic areas (e.g., right and left hemisphere V1), areas neighboring homotopic regions (e.g., right hemisphere V1 and left hemisphere V2), and areas distal to homotopic regions (e.g., right hemisphere V1 and left hemisphere V3). The strength of correlation varied as a function of areal group (2-way ANOVA with monkey and areal grouping as factors; ps<0.01; *Figure 2* legend). Across all 16 areas tested, pairwise correlations were strongest between homotopic cortical areas, followed by cortical areas neighboring homotopic regions, and then cortical areas distal, non-neighboring to

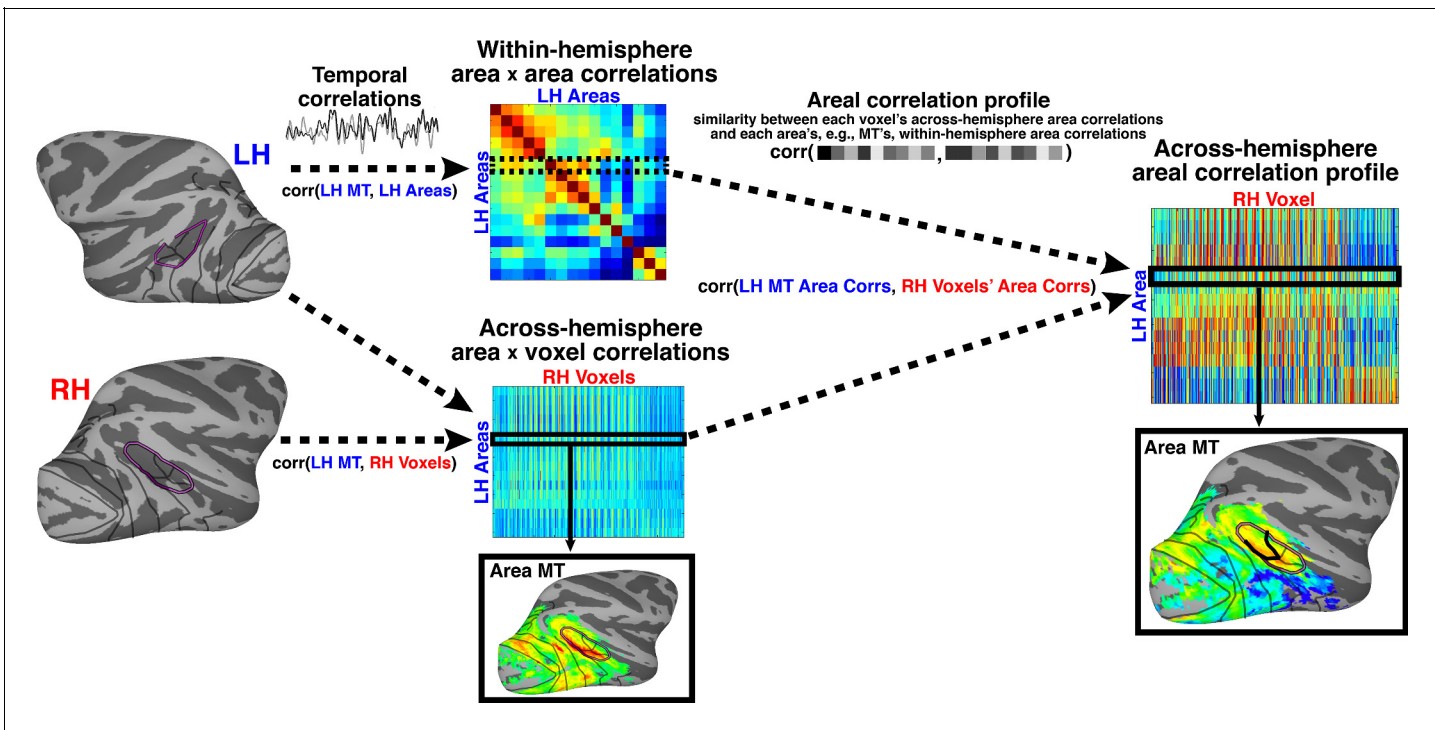

**Figure 3.** Pipeline of correlation analysis. The two-step analysis consisted of first measuring temporal correlations followed by the areal correlation profile. First, within-hemisphere pairwise temporal correlations were calculated between 16 retinotopic seed cortical areas for each hemisphere. The resulting matrix (labeled 'Within hemisphere area x area correlations') is, by definition, symmetric, with each row or column corresponding to correlations between one seed cortical area and all other seed areas. Figure example is given for left hemisphere. Temporal correlations were also computed between each cortical seed area and voxels in the opposite hemisphere. Figure example is for left hemisphere seed areas and right hemisphere voxels. Each row of this matrix (labeled 'Across-hemisphere area x voxel correlations') can be visualized as the spatial map of correlated activity for a given seed area (e.g., MT) with every voxel in the contralateral hemisphere. This is the typical approach for voxel-wise, seed-based correlation analyses (*Biswal et al., 1995*). Next, the within-hemisphere area x area matrix was correlated with the across-hemisphere area x voxel matrix of the contralateral hemisphere. This yields a matrix (labeled 'Across-hemisphere areal correlation profile') of similarity measurements between the profile of each seed area's within-hemisphere correlations with all other areas and the correlation profile of each voxel in the contralateral hemisphere with all other areas. Each row of this matrix can be visualized as a spatial map of similarity between a given area's (e.g., MT) within-hemisphere correlations and each contralateral voxel's across hemisphere area correlations. i.e., illustrating which voxels have an across-hemisphere area correlation pattern similar to contralateral MT. Code for analysis available on GitHub (*Arcaro, 2017*) A copy is archived on https://github.com/elifesciences-publications/AreaProfileCorrelation.

homotopic regions (two-tailed, unpaired t-tests; Homotopic vs. Adjacent: t(134) = 3.96; Adjacent vs. Distal: t(478) > 10.27; ps<0.001). We verified that these correlations could not be attributed to biases in instrumental sampling or preprocessing by performing a control analysis where we simulated data with approximately the intrinsic spatial smoothness of fMRI data (*Smirnakis et al., 2007*) and then passed these data through our analysis pipeline (Materials and methods: *Simulation of correlation due to instrumentation and analyses*). Across hemisphere correlations from these simulated data were not significantly different from 0 (*Figure 2b*; grey box plots). This control analysis also revealed that such potential biases affected only within-hemisphere correlations for neighboring areas, and even then could not fully account for the observed correlation structure.

Areal specificity was also observed at the individual voxel-level. As a first pass, we performed a standard seed-based correlation analysis where we correlated the mean activity of each seed area with the activity of each voxel in the opposite hemisphere. We observed significant across-hemisphere voxel-wise correlations for all 16 seed areas in each monkey (r > 0.15; t > 5.66; p<0.0001, FDR-corrected). Voxel-wise correlations were almost exclusively within the extent of visual cortex and peak correlations were within each seed's homotopic cortical area (e.g., across-hemisphere correlations for area MT in left hemisphere were strongest with right-hemisphere voxels in MT; *Figure 3*).

To increase the spatial specificity of the correlation analysis, we used an approach that, in addition to comparing pairwise correlations between individual voxels and seed areas, also factors in the similarity structure between seed areas (Materials and methods: *Areal correlation profile*). To do this, we first measured the (temporal) correlations between all seed areas within the same hemisphere; which yielded a measure of the similarity between each seed area's mean activity and all other seed areas' mean activity (*Figure 3*). This is comparable to the analysis in the previous paragraph (*Figure 2*), but computed within, not across, hemispheres. As expected, the mean activity for many seed areas was correlated (rs > 0.15; ts > 5.66; ps<0.0001, FDR-corrected). Temporal correlations were also calculated between the mean activity of each seed area and the activity of every voxel in the contralateral hemisphere. Then, for each seed area, we compared the within-hemisphere (area x area) correlation measure with the across-hemisphere (area x voxel) correlation measure for voxels in the opposite hemisphere. This areal correlation profile approach yielded a measure of similarity between each voxel and each seed area that factors in the similarity between areal seeds, and

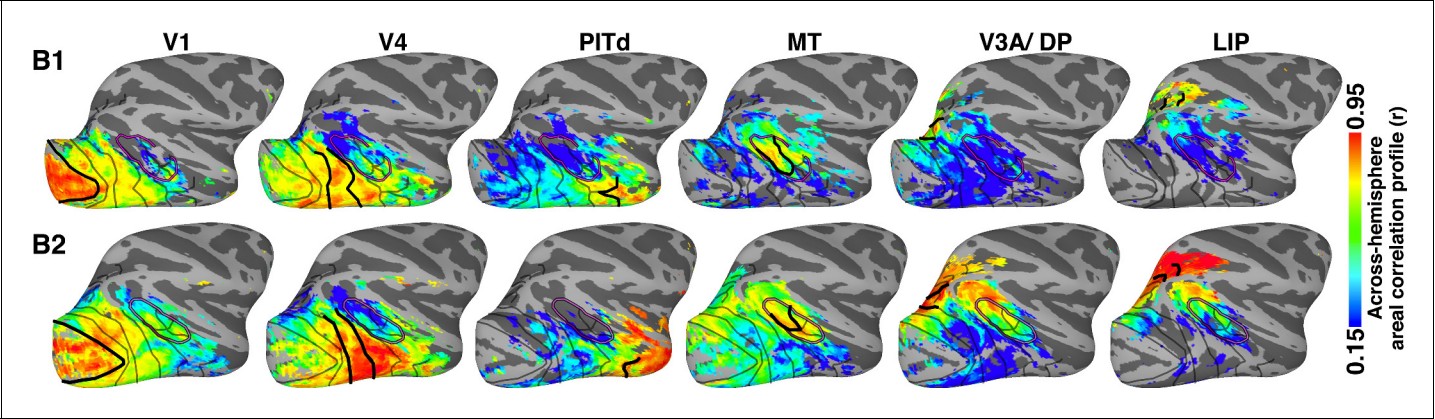

**Figure 4.** Across-hemisphere areal correlation profile maps for retinotopic seed areas in occipital (V1 and V4), middle temporal (MT), posterior IT (PIT), dorsal extrastriate (V3A/DP), and lateral intraparietal cortex (LIP) in the right hemispheres of newborn monkeys B1 and B2. Thick black lines mark the borders of the area whose counterpart in the contralateral hemisphere served as the seed; areas were identified by retinotopic mapping in the same monkey at >1.5 years old. Data threshold on the initial across-hemisphere temporal correlations (r > 0.15; t > 5.66; p<0.0001, FDR-corrected). See *Figure 4—figure supplement 1* for (a) comparison of contralateral correlations for both hemispheres in both monkeys and for (b) within-hemisphere correlation maps. Thinner black lines mark what we could later identify as the borders between visual areas. Purple line corresponds to what we could later identify as the peripheral eccentricity ridge of the MT cluster comprising areas MT, MST, FST, V4t; (*Kolster et al., 2009*).

The following figure supplement is available for figure 4:

**Figure supplement 1.** Across- and within-hemisphere similarity maps.

thus enhances the specificity of correlation patterns. To ensure that a given voxel's activity was similar to that of the seed area, the initial across-hemisphere temporal correlations were used as the threshold (r > 0.15; t > 5.66; p<0.0001, FDR-corrected). Across the 16 seed cortical areas, across-hemisphere areal correlation profiles were strongest for voxels within the homotopic area, with areal correlation profile strength dropping off with distance (*Figure 4*; *Figure 4—figure supplement 1a*). Within-hemisphere areal correlation profile maps were comparable to the across-hemisphere maps for matched seed areas (*Figure 4—figure supplement 1b*). Taken together, these areal correlation profile maps further demonstrated that arealization of the entire visual hierarchy already existed within the first weeks after birth.

## Organization of the visual pathway into dorsal and ventral streams in neonates

Community structure analyses of neonatal correlation patterns revealed a large-scale organization of the visual system into dorsal and ventral visual subdivisions, as is found in adult monkeys (*Ungerleider and Mishkin, 1982*). Multidimensional scaling of the neonatal across-hemisphere (temporal) correlations between 16 visual cortical areas (*Figure 2a*), revealed the two principle dimensions of structure: posterior-anterior and dorsal-ventral (*Figure 5a*). This indicates that the cortical location of an area was a good predictor of its connectivity; this is similar to what has been found in adult monkeys using within-hemisphere anatomical connections (*Young, 1992*; *Markov et al., 2011*). Spectral community structure analysis of neonatal correlations partitioned these 16 cortical areas into 3 groups: (1) occipital areas V1, V2, and V3 (green dots), (2) ventral temporal areas V4, V4A, OTd, PITv, PITd, and OTS (red dots), and (3) dorsal areas V3A/DP, CIP, LIP, as well as the MT

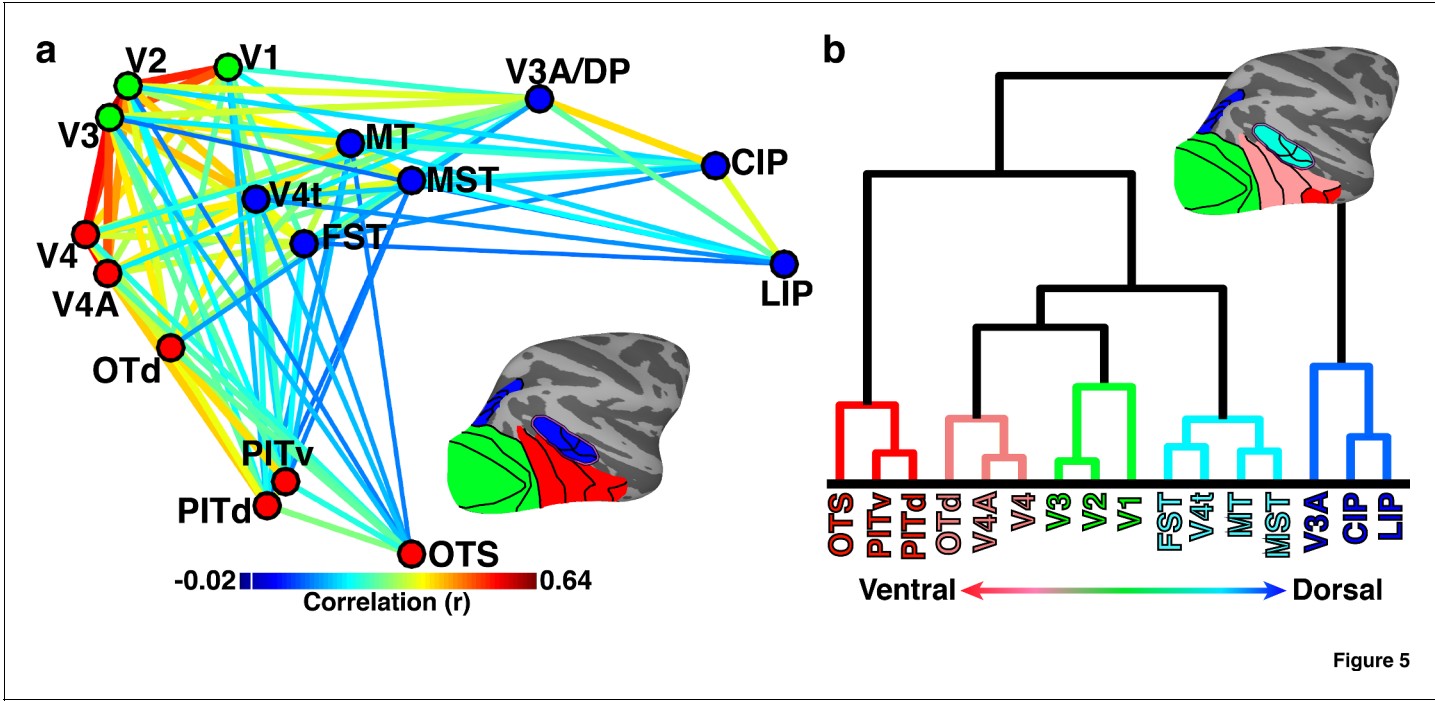

**Figure 5.** Hierarchical organization of dorsal and ventral visual pathways in newborns. (a) Multidimensional scaling (MDS) and community segmentation on pairwise contralateral correlations of the mean signals between 16 areal seeds (See *Figure 2* for seed correlation matrix). Data were collapsed across hemispheres prior to MDS. Average monkey pairwise correlations (r) >0.15 are visualized with correlation strength indicated by line color (ts > 5.66; ps<0.0001, FDR-corrected). Dots are color-coded based on community clustering. Areas showed a modest degree of modularity (Q = 0.11). Colorbar range scaled to min and max of matrix in *Figure 2a*. White line in colorbar indicates 0. (b) Hierarchical clustering of pairwise contralateral correlations. Leaf (area) ordering of the dendrogram maximized similarities between adjacent leaves (areas). The functional distances between areas were well captured by the hierarchical dendrogram (cophenetic correlation = 0.83). Lines are color-coded based on hierarchical cluster. For both sets of data, the inset lateral surfaces illustrate the anatomical location and cluster membership of each area by matching color code. OTS (not shown) is located on the ventral surface within a region of the occipitotemporal sulcus medial to area PITv.

cluster (blue dots). Consistent with the organization found in adult monkeys, MT was functionally grouped with dorsal stream areas despite its anatomical location in temporal cortex and proximity to ventral stream areas V4, OTd, and PIT (*Ungerleider and Mishkin, 1982*). Occipital cortical areas were correlated with both ventral and dorsal groups. Correlations between dorsal and ventral groups were also present, though relatively weak compared to correlations within groups. This analysis thus yielded a clear distinction, at birth, between dorsal and ventral visual pathways, which is similar to the known organization of the adult visual system.

Hierarchical cluster analysis revealed nested layers within the functional segregation of dorsal and ventral pathways at birth (*Figure 5b*). V1 was situated in the middle of the dendrogram with one set of areas (left) extending into inferior temporal cortex and other areas (right) extending into parietal cortex. Hierarchical clustering resulted in a finer-grained parcellation into five clusters: posterior occipital areas V1, V2, and V3 (green), anterior occipital areas V4, V4A, and OTd (peach), inferior temporal areas PITd, PITv, and OTS (red), the MT cluster (cyan), and dorsal areas V3A, CIP, LIP (blue). The highest tier of the tree distinguished dorsal occipital and parietal cluster from occipital and temporal clusters. The next tier distinguished the occipital and MT clusters from inferior temporal cluster. Subsequent tiers differentiated the occipital clusters from the MT cluster and posterior occipital cluster from anterior occipital cluster. Taken together, the multidimensional scaling and hierarchical clustering revealed the presence, at birth, of the well-established (in adults) broad differentiation between dorsal and ventral visual pathways as well as a hierarchical organization within each pathway.

## Retinotopic organization in neonates

An even finer level of differentiation within individual neonatal visual areas was apparent as a series of retinotopic representations that could be inferred from correlation patterns. First, within-hemisphere correlations were assessed between posterior areas V1, V2, V3, V4, and V4A as a function of their dorsal and ventral quadrants, representing lower and upper visual fields, respectively. Within-hemisphere correlations between dorsal or ventral quadrant pairs (e.g., dorsal V1 and dorsal V2) were stronger than correlations between non-matched quadrant pairs (2-way ANOVA; main effects of quadrant group: $F(1,156) = 45.1$, $p<0.0001$; marginal effect of monkey $F(1,156)=3.63$, $p<0.06$; no interaction $p>0.10$) even when excluding correlations between adjacent areas (e.g., V1 and V2), which mitigates the influence of fMRI signal spread (2-way ANOVA; main effect of quadrant group: $F(1,92) = 12.8$, $p<0.001$ and monkey; main effect of monkey: $F(1,92) = 4.29$, $p<0.05$; no interaction $p>0.10$). Though correlations from simulated data were much weaker than the real data (Materials and methods: Simulation of correlation due to instrumentation and analyses), within-hemisphere correlations between quadrants could be partly biased by the intrinsic spread of the fMRI signal. To further rule out the influence of cortical proximity on retinotopic correlations, across-hemisphere correlations were assessed between posterior areas V1, V2, V3, V4, and V4A as a function of their dorsal and ventral quadrants. The mean signals of homotopic quadrants across hemispheres were significantly correlated in each monkey ($ts(9)>17.21$; $ps<0.001$). The checkerboard pattern of the correlation matrix in *Figure 6a* indicates that the homotopic correlations were stronger for corresponding dorsal or ventral quadrants than for non-corresponding quadrants (*Figure 6b*; 2-way ANOVA with monkey and quadrant group as factors; $p<0.0001$). Across-hemisphere correlations between non-homotopic regions were also greater for corresponding dorsal or ventral quadrants (e.g., dorsal V1 right hemisphere to dorsal V3 left hemisphere) than for non-corresponding quadrants (2-way ANOVA; main effect of quadrant group: $F(1,92) = 10.87$, $p<0.0014$; no main effect of monkey, $p>0.10$; no interaction between monkey and quadrant group, $p>0.10$). Because dorsal:dorsal and ventral:ventral correlations were stronger than dorsal:ventral correlations even across areas (e.g., V1 to V3), these correlations likely reflect the underlying topographic organization within each area, not solely mirror symmetrical point-to-point connections between hemispheres. Even though right and left hemispheres represent left and right visual fields, respectively, callosal connections linking neurons with overlapping receptive fields at or near the vertical meridian (*Dehay et al., 1986*) can account for correlated activity between hemispheres in adults (*Heinzle et al., 2011*). We again verified that these correlations could not be attributed to biases in instrumental sampling or preprocessing with the simulated data described in *Figure 2* (Materials and methods: *Simulation of correlation due to instrumentation and analyses*). Across hemisphere correlations from these simulated data were not significantly different from 0 (*Figure 6b*; grey box plots). These data demonstrate a coarse

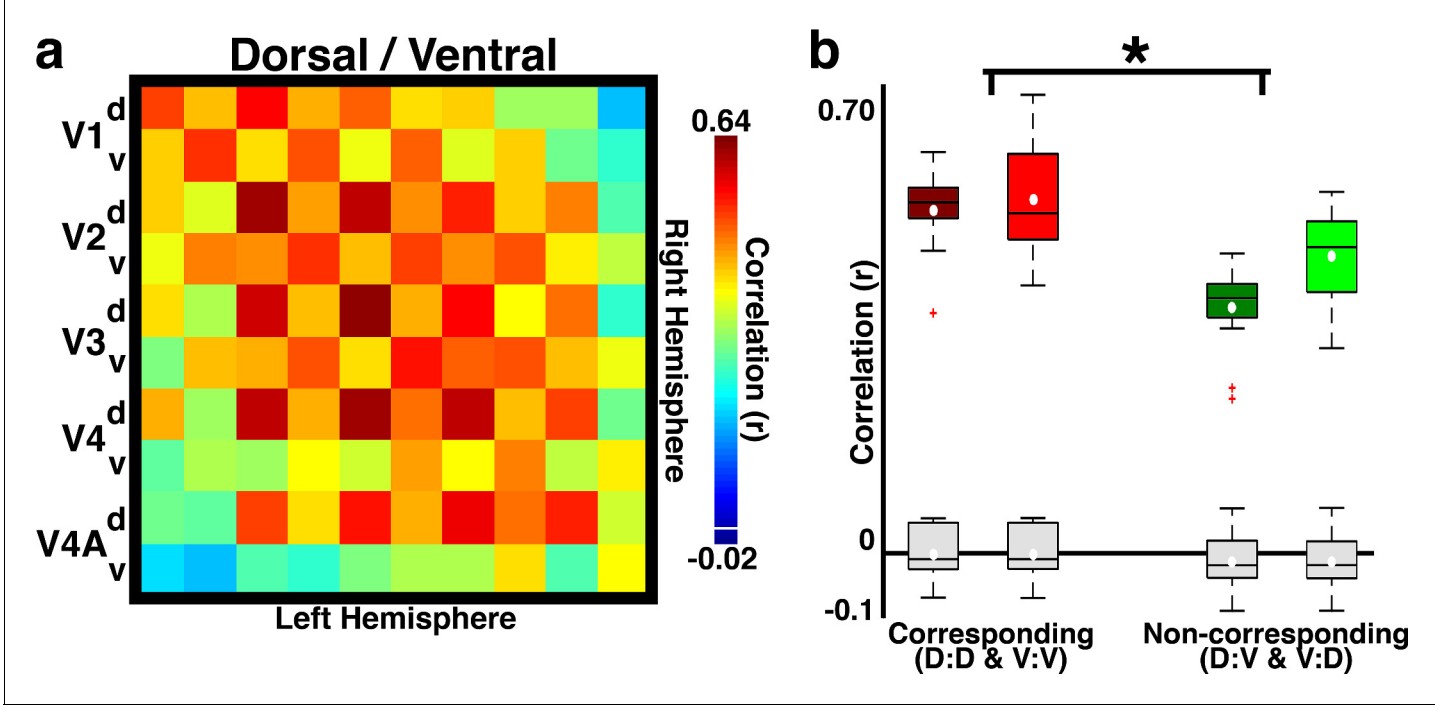

**Figure 6.** Temporal correlations distinguish upper and lower visual field representations. (**a**) Across-hemisphere correlation matrix for dorsal/ventral quadrants of areas V1, V2, V3, V4, and V4A. The checkerboard pattern reveals dorsal vs. ventral visual field specificity for within- and between-area correlations. Colorbar range scaled to min and max of matrix in *Figure 2a*. White line in colorbar indicates 0. (**b**) Box plot for corresponding and non-corresponding dorsal vs. ventral quadrants for homotopic areas V1, V2, V3, V4, and V4A in B1 (dark boxes) and B2 (light boxes). Mean (white circle), median (black horizontal line), interquartile (whiskers), and outliers (red crosses) are presented. Correlations were strongest between corresponding (vs. non-corresponding) quadrants (2-way ANOVA with monkey and quadrant group as factors; main effect of quadrant group: $F(1,52) = 33.17$, $p<0.0001$; main effect of monkey: $F(1,52) = 5.47$, $p<0.05$; no interaction between monkey and quadrant group, $p>0.1$). Comparable effects were found for data averaged across hemisphere. Grey box plots show the correlation expected from any biases in our fMRI acquisition or analyses (Materials and methods: Simulation of correlation due to instrumentation and analyses).

topographic organization within individual areas of newborn monkeys that likely reflects retinotopic organization.

We then used seed-based correlation analysis to reveal a finer-scale retinotopic organization throughout visual cortex in neonates. The IC analysis (*Figure 1—figure supplement 2*) had revealed a broad foveal vs. peripheral distinction, and the checkerboard pattern of correlations between dorsal vs. ventral subdivisions of individual occipital areas (*Figure 6*) indicated a lower vs. upper visual-field specificity. We then asked how detailed this retinotopic organization was across visual cortex. At this age, we could not perform conventional retinotopic mapping. Instead, we evaluated whether within-hemisphere correlated activity patterns at this early age reflected the retinotopic organization measured in the same monkeys much later in life (*Figure 1—figure supplement 1*). Area V1 was segmented into eight eccentricity bands covering the central 10° of V1's retinotopic map. We computed whole-brain temporal correlations between the activity of each voxel and the mean activity for each V1 eccentricity band (*Figure 7*; *Figure 7—figure supplement 1*). To identify each voxel's preferred eccentricity, its correlation coefficients for each eccentricity bin were fit with a Gaussian curve and the peak of the curve was identified (See Materials and methods and *Figure 7—figure supplement 1*).

The resulting eccentricity correlation maps obtained in neonates were predictive of the eccentricity organization derived by direct retinotopic mapping in the same monkeys when they were older and able to fixate continuously with high accuracy (within 1°) for several minutes (*Figure 7*). Specifically, lateral occipital cortex had foveal correlations in neonates that extended anteriorly along the lower lip of the STS. Moving towards the medial occipital surface, eccentricity correlations gradually shifted towards the periphery. A region of foveal correlation, distinct from the foveal confluence of

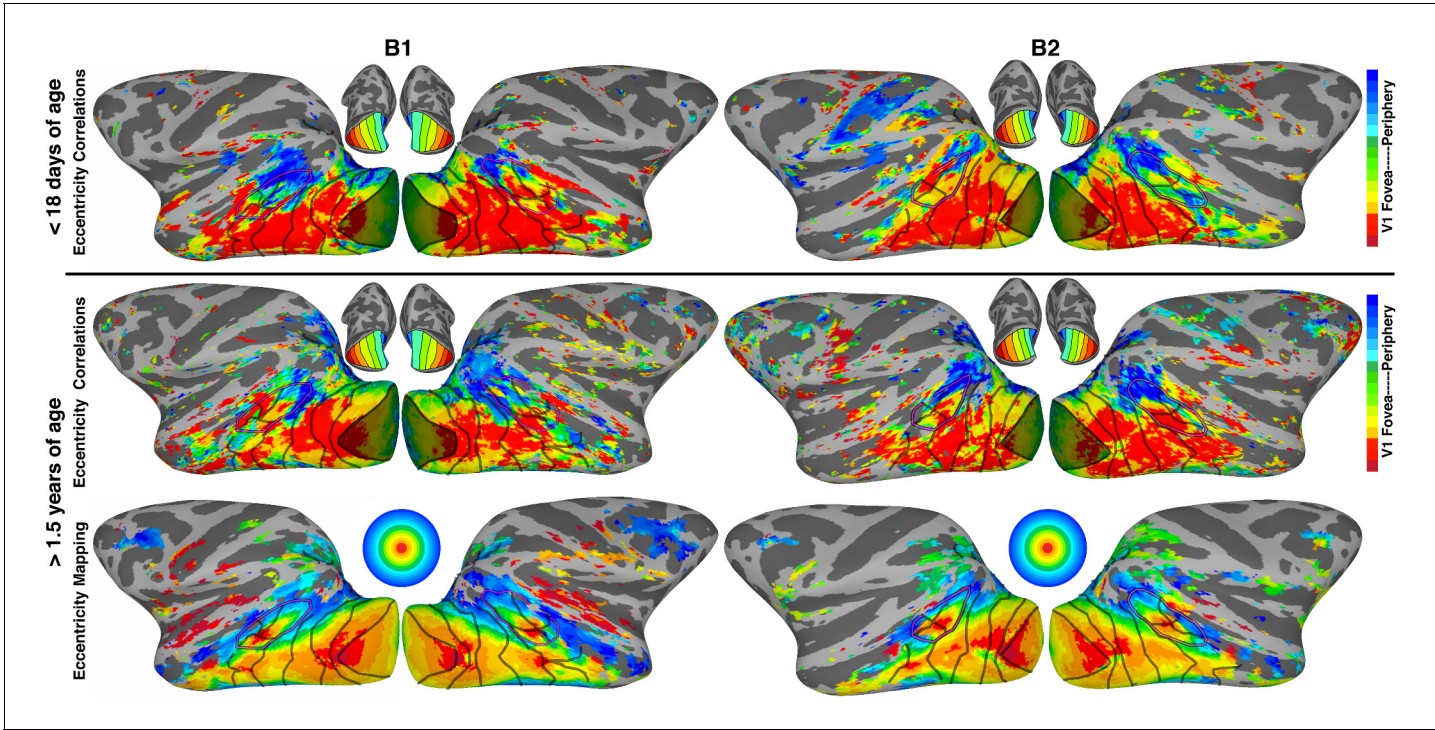

**Figure 7.** Retinotopically-specific correlated activity across ipsilateral visual cortex. Spatial maps color-coded according to peak correlations with ipsilateral V1 eccentricity bands (top) in two neonatal monkeys < 18 days postnatal and (middle) in the same monkeys at ~3 years. Shaded area indicates V1 seeds. (bottom) Eccentricity maps (by conventional retinotopic mapping) in same monkeys at >1.5 years old. Excluding V1, the mean absolute deviation between (top) eccentricity correlations in neonates and (bottom) the eccentricity maps from retinotopic mapping was 2.2° (1.4°, 2.4°, 1.9°, and 2.1° for occipital areas V1-V4, MT cluster, inferior temporal areas, and dorsal/parietal areas, respectively). Excluding V1, the mean absolute deviation between eccentricity correlations (top) in neonates and (middle) in juveniles was 2.0° (1.8°, 2.5°, 1.7°, and 2.5° for occipital areas, MT cluster, inferior temporal, and dorsal/parietal areas, respectively). Excluding V1, the mean absolute deviation between (middle) eccentricity correlations in juveniles and (bottom) the eccentricity maps from retinotopic mapping was 1.4° (1.4°, 1.3°, 1.4°, and 1.3° for occipital, MT cluster, inferior temporal, and dorsal/parietal areas, respectively). Maps show lateral surface of the brain; see *Figure 1* for additional conventions.

The following figure supplement is available for figure 7:

**Figure supplement 1.** Illustration of curve fitting of V1 eccentricity seed correlations.

early visual cortex, was present within the fundus of the STS, corresponding to what later could be identified as the fovea of the MT cluster. Parafoveal and peripheral correlations encircled this fovea, especially in the dorsal-most portions of the STS near the prelunate gyrus and crown of the IPS; consistent with the retinotopic organization identified when the monkeys were older. Correlations were mostly sub-threshold within dorsal extrastriate cortex near V3A as well as within the IPS, but eccentricity correlations generally represented the periphery, also consistent with subsequent retinotopic mapping. Excluding V1 (i.e., the seed area), the mean absolute deviation between eccentricity correlations in newborns and the eccentricity measurements from retinotopic mapping after 1.5 years of age was 2.2° across all retinotopic areas, in both hemispheres, in both monkeys. Given the close spatial proximity between V1 and V2, some correlation should be expected due to spatial spread of the fMRI signal (Materials and methods: *Simulation of correlation due to instrumentation and analyses*). However, this influence was minimal. Actual V1 correlations with V2 and V3 were approximately 2x and 5x, respectively, greater than estimated correlations attributable to spatial spread of the fMRI signal. Anterior to V3, correlations with V1 in our simulation were negligible. The neonate eccentricity correlation maps were also similar to correlation maps obtained in the same monkeys as juveniles (~3 years of age), indicating that retinotopic correlation patterns are present through development. Excluding V1, the mean absolute deviation between eccentricity correlations at newborn and

juvenile ages was 2.0° across retinotopic areas, in both hemispheres, in both monkeys. Juvenile eccentricity correlations were more similar to the eccentricity mapping (mean deviation = 1.4°) than to the neonate eccentricity correlations, potentially indicating refinement of retinotopic maps over development. However, these differences might reflect non-biological variance (e.g., the precision of anatomical registration and proximity of coil placement due to brain size differences across ages). These data indicate that extensive retinotopic organization across both early and higher visual cortex was already present within the first weeks of life.

These early eccentricity maps extended into regions of IT cortex that in adults are organized into domains selectively responsive to different object categories, like faces (*Tsao et al., 2003*; *Pinsk et al., 2009*; *Bell et al., 2009*). At this young age, however, regions of IT selectively responsive to face and non-face stimuli (measured with fMRI) were not differentiable in IT cortex (*Livingstone et al., 2017*). Face patches would not emerge until about 200 days in these monkeys (*Figure 8a*, thick black outline). Yet foveal biases were identified in this neonatal data along the lower lip of the STS, including a region that would eventually be identifiable as the middle face patch (ML), consistent with previous studies that compared eccentricity and face patch organization in adult monkeys (*Janssens et al., 2014*; *Rajimehr et al., 2014*; *Lafer-Sousa and Conway, 2013*). In neonates, the eccentricity correlations within what would become area ML were mostly foveal, though midfield and peripheral representations were identified on the medial edge within the fundus in 3 of 4 hemispheres. This progression of eccentricity representations was consistent with eccentricity representations from subsequent mapping in these same monkeys at >1.5 years of age, and with a previous study in adult monkeys (*Janssens et al., 2014*). We further tested the retinotopic bias of these precursor-face-patches in neonates by correlating the mean signal of the region that would become ML with the mean signal of each V1 eccentricity bin. Correlations of the region that would become ML were strongest with the most foveal V1 bin and diminished towards the periphery (*Figure 8b*). These correlations could not be accounted for by instrumentation or analysis procedures (*Figure 8b*; red line). The pattern of V1 eccentricity correlations in neonates paralleled the distribution of eccentricity representations in ML from subsequent retinotopic mapping in these same monkeys at >1.5 years; both of which showed an overall foveal bias (*Figure 8b and c*).

Though eccentricity representations were weak in the most anterior parts of IT, the anterior face patch (AL) showed a bimodal retinotopic correlation pattern, with the strongest correlations at foveal- and peripheral-most V1 bins, consistent with the sub-threshold distribution of eccentricity representations mapped >1.5 years later (*Figure 8—figure supplement 1*). Eccentricity correlations in the fundal face patches (MF and AF) were not investigated because eccentricity representations in our mapping varied in the STS fundus between monkeys, and the retinotopic organization, if any, in the fundus remains unclear (*Janssens et al., 2014*). Together, these data indicate that retinotopic organization of what would become category-selective regions in IT cortex was already established within the first weeks after birth.

Though neonatal correlation patterns distinguished retinotopic areas within IT (*Figures 2* and *4*) and neonatal ICA analysis revealed a broad IT component (*Figure 1—figure supplement 2*), neither analysis revealed a face-patch organization in neonates. These data are consistent with a recent study from our lab showing that task-evoked face-selective clusters in IT do not appear until about 200 days of age (*Livingstone et al., 2017*). The failure to find a face-selective IC early in development cannot be ascribed to insensitivity of this analysis. We conducted ICA on several data sets at different ages in which the monkeys had viewed blocks of faces and other images. After 200 days of age both IC analysis and univariate analysis revealed face patches; before 200 days neither showed face-patch organization. (*Figure 8—figure supplement 2*) (*Livingstone et al., 2017*). In these older datasets, foveal and peripheral ICs were also identified, further demonstrating the stability of retinotopic organization through development. For datasets from monkeys older than 200 days, IC components comprising the face patch system also included foveal and parafoveal (~1°−6°) regions of occipital cortex, demonstrating that the link between retinotopy and category selective domains is consistent through development. Taken together, these data indicate that the retinotopic areal organization in inferior temporal cortex precedes category selectivity.

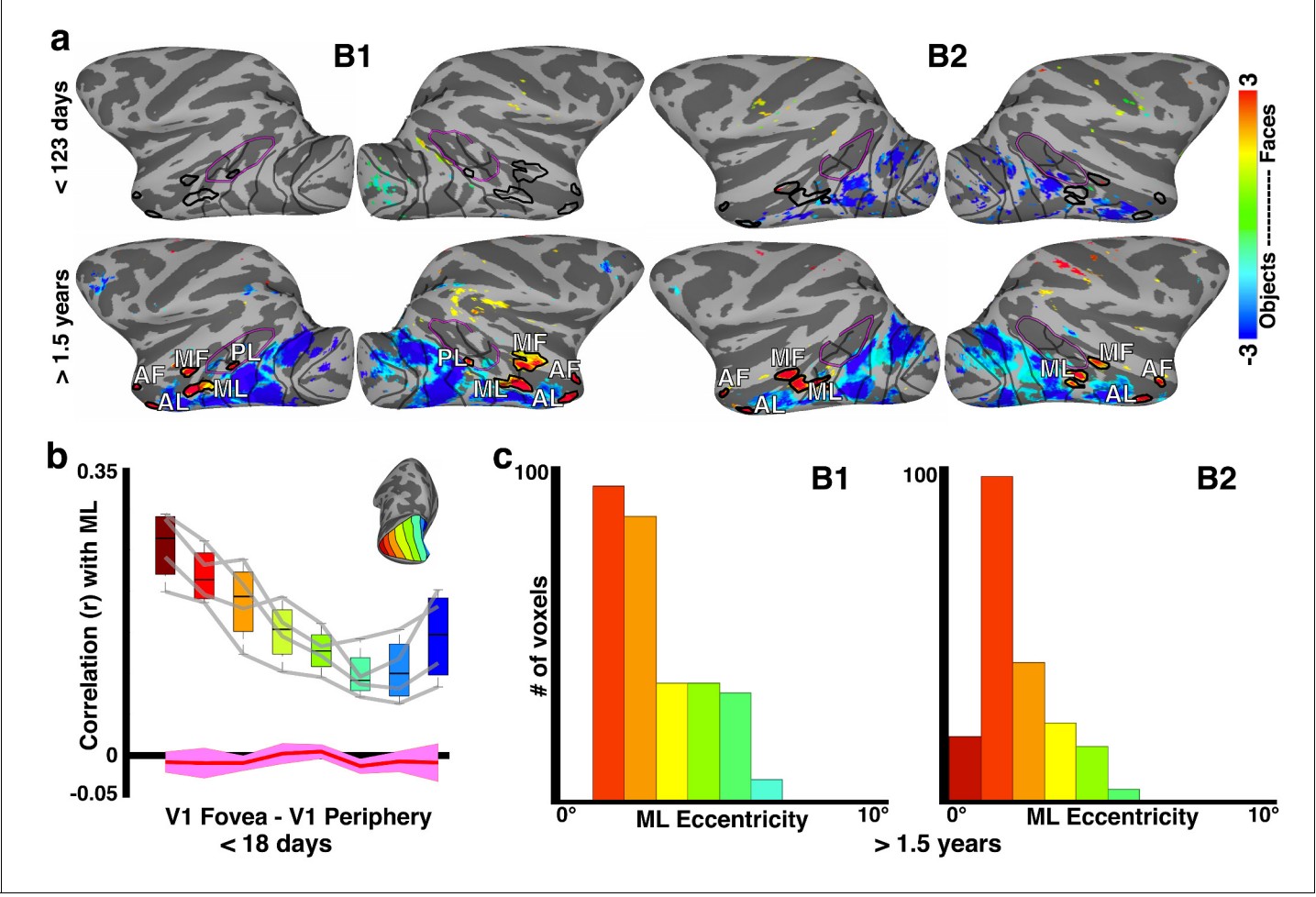

**Figure 8.** Relationship between correlation-based retinotopy and precursor-face-patches in newborns. (**a**) Faces-minus-objects contrast maps for monkeys B1 and B2 at <123 days and >1.5 years. In each animal, the contrast maps for <123 days summarizes data across four scan sessions reported individually in *Livingstone et al. (2017)*. Average beta contrasts are shown for voxels that were significant in at least two scan sessions (p<0.05, FDR-corrected). The contrast maps for >1.5 years are from a single session in each animal when the face patches were stable and robust (p<0.05, FDR corrected). Thick black outlines illustrate the extent of face-selective regions in IT as defined at >1.5 years by the faces-minus-objects contrast; these outlines were overlaid onto the neonatal data at <123 days. (**b**) Box plots of correlations between the precursor-middle face patch and V1 eccentricity bands in neonates < 18 days old averaged across hemispheres and monkeys. Data from individual hemispheres are plotted as gray lines. Red line illustrates correlations expected from instrumentation noise and preprocessing. (**c**) Histogram of eccentricity representations averaged across all voxels in the middle face patch (ML) and both monkeys >1.5 years of age.

The following figure supplements are available for figure 8:

**Figure supplement 1.** Sub-threshold retinotopic correlations with anterior face patch, AL.

**Figure supplement 2.** Data-driven spatial ICA maps in four additional data sets.

## Low-level tuning correlates with retinotopic organization

In adult monkeys, receptive-field size and preferred stimulus size scale with eccentricity (*Hubel and Wiesel, 1974*); smaller receptive fields in the fovea prefer high spatial frequencies and short or curved contours; peripheral receptive fields are tuned to lower spatial frequencies and longer straight contours (*Hubel and Wiesel, 1965*; *De Valois et al., 1982*). Therefore, in adults, maps of both spatial frequency and curvature tuning correlate with eccentricity maps (*Ponce et al., 2017*; *Srihasam et al., 2014*; *Henriksson et al., 2008*). To see whether the eccentricity organization we found in neonates carried with it a similar selectivity for stimulus scale, we performed spatial

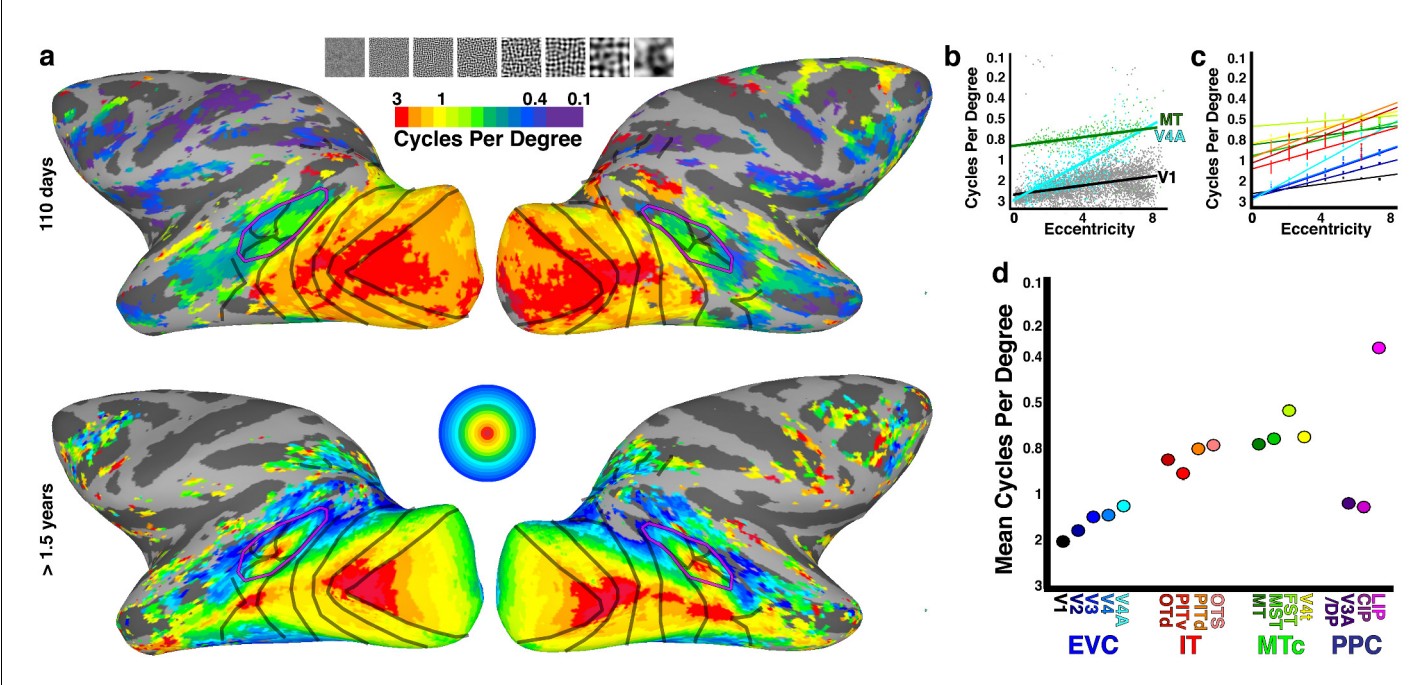

**Figure 9.** Spatial tuning correlates with retinotopic organization. (**a**, top) Cortical surface maps of preferred spatial frequency in monkey B3 at 110 days. Data threshold at p<0.001. (**a**, bottom) Cortical surface map of average eccentricity organization calculated from four other monkeys at >1.5 years. Data threshold such that only voxels with p<0.001 in at least 3 out of 4 monkeys are displayed. See *Figure 1* for additional conventions. (**b**) Scatter plot of preferred spatial frequency (i.e., cycles per degree) in every voxel in three visual areas vs. eccentricity; lines show linear fit. (**c**) Linear fits of preferred spatial frequency vs. eccentricity for areas V1-V4A, MT, MST, FST, V4t, OTd, PITv, and PITd as indicated by corresponding color in panel d. Areas CIP1/2, V3A/DP, OTS1/2 and LIP were not included due to two or more eccentricity bins lacking any voxels. (**d**) Mean preferred spatial frequency for each retinotopic area.

frequency mapping in one infant monkey at 110 days old, well after significant visual responses had appeared with fMRI (*Figure 1*), but prior to the emergence of face-selective clusters in IT (*Figure 8a*). At this age, the monkey could fixate on the stimulus screen, but not with the accuracy and consistency required for retinotopic mapping. The spatial frequency was constant across each 20° image, and therefore it was necessary only that the monkey look at the screen. We used a phase-encoded paradigm (*Henriksson et al., 2008*) to map tuning across eight spatial frequencies (See Materials and methods: Spatial frequency mapping). As in adults, preferred spatial frequency varied as a function of both visual hierarchy and eccentricity (*Figure 9a*). In particular, in each area, foveal regions preferred the highest spatial frequencies and peripheral regions lower spatial frequencies (*Figure 9b–d*). In addition, the mean preferred spatial frequency decreased moving up the hierarchy, in both dorsal and ventral pathways (*Figure 9d*). Thus, although we could not do spatial frequency mapping in neonatal monkeys, we nevertheless did observe a correlation between eccentricity and spatial frequency in infants that were slightly older but still prior to the emergence of category selectivity. We therefore assume that the presence of retinotopy, as measured by correlation patterns in the neonates, implies a corresponding early organization for receptive field scale.

Because curvature selectivity varies with eccentricity in adult monkeys (*Ponce et al., 2017*; *Srihasam et al., 2014*), we asked whether an organization for this low-level shape feature was present in infant monkeys. Responses to curved and rectilinear stimuli were measured in monkey B4 at 186 days (*Figure 10*). This is around the age when face-selective clusters first emerge (*Livingstone et al., 2017*). Similar to spatial frequency, curvature tuning varied as a function of both area and eccentricity (*Figure 10b–e*). Only V1-V4A, OTd, and PITd showed significant selectivity for curvature, and in each of these areas foveal regions preferred curved stimuli more than rectilinear, and peripheral regions either showed no preference or preferred rectilinear stimuli (*Figure 10c–d*). Therefore an organization for curvature tuning is present at least as early as the time in development

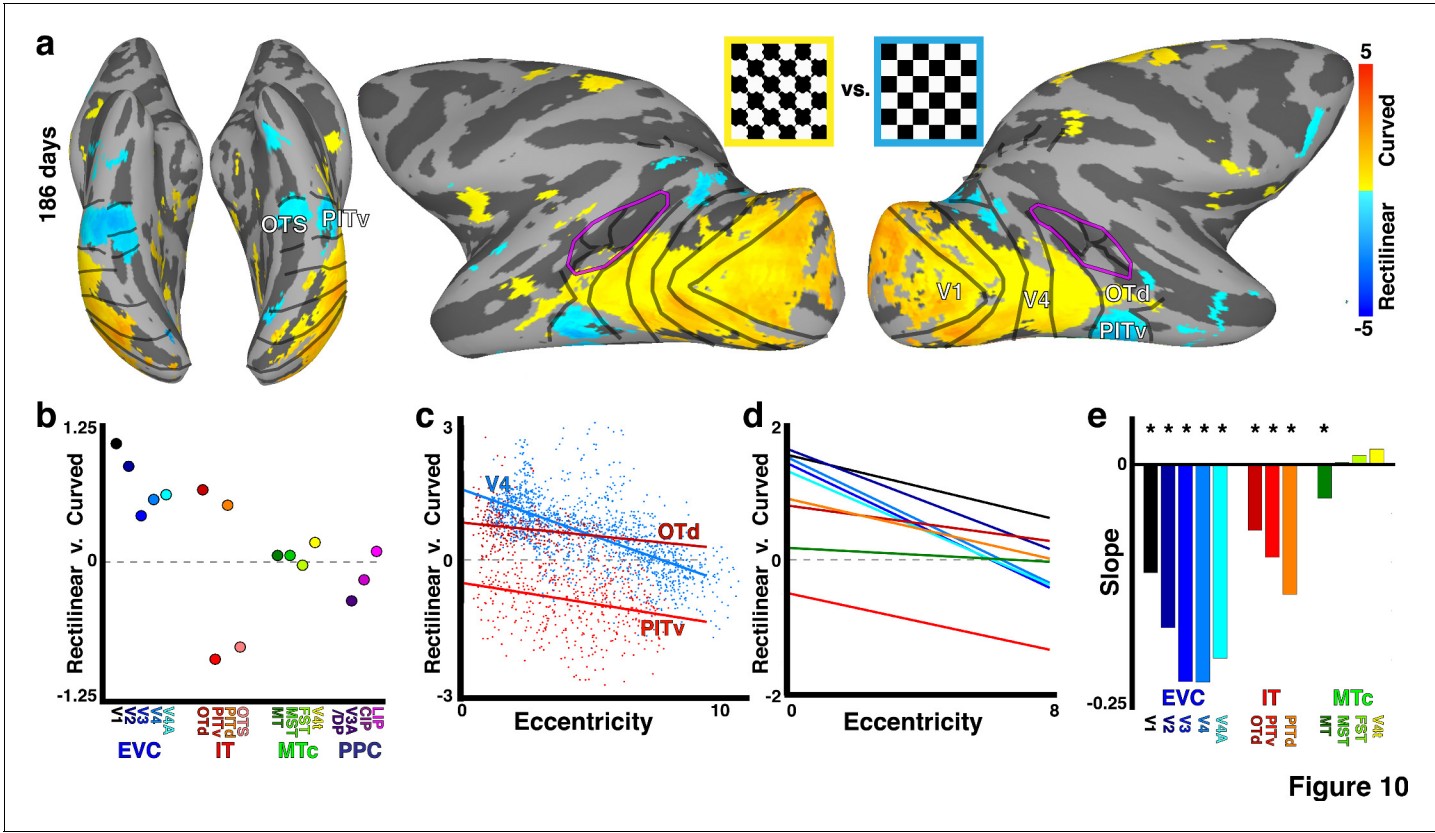

**Figure 10.** Low-level shape selectivity correlated with retinotopic organization. (**a**) Cortical surface map of curved-minus-rectilinear contrast in monkey B4 at 186 days. Data threshold at p<0.05 FDR corrected. (**b**) Mean response contrast to curved-minus-rectilinear stimuli for each retinotopic area. (**c**) Scatter plot of the curved-minus-rectilinear contrast as a function of eccentricity for areas V4, OTd, and PITv; lines show linear fit. (**d**) Linear fits of the curved-minus-rectilinear contrast vs. eccentricity for areas V1-V4A, MT, MST, FST, V4t, OTd, PITv, and PITd as indicated by corresponding color in panel e. Areas CIP1/2, V3A/DP, OTS1/2, and LIP were not included in the linear fits due to two or more eccentricity bins lacking any voxels. Eccentricity bins comprising less than five voxels were excluded from the plot. (**e**) Slopes of linear fits (p<0.05 FDR corrected as indicated by asterisk).

that face clusters emerge. Area PITv showed a correlation between curvature and eccentricity, but in addition showed stronger average selectivity for rectilinear stimuli. V3A/DP also showed stronger average selectivity for rectilinear stimuli. These two regions correspond to scene-selective cortex (*Nasr et al., 2011*), and have been shown to prefer rectilinear stimuli also in adults (*Yue et al., 2014*). OTS (combined areas OTS1 and OTS2) also showed stronger average selectivity for rectilinear stimuli. OTS1 and OTS2 are located in a region of the occipitotemporal sulcus that likely corresponds to the scene area LPP (*Arcaro and Livingstone, 2017*), which also prefers rectilinear stimuli in adults (*Kornblith et al., 2013*). Thus, it seems likely that an organization for low-level shape selectivity precedes category selectivity, and contributes to the formation of category-selective domains.

## Discussion

The visual system was found to be organized to a remarkable degree in newborn macaques as young as 10 days postnatal. Visual cortex was functionally distinct from other sensory and non-sensory regions. Across-hemisphere correlations revealed arealization and a hierarchical organization that was similar to that seen in adults. Correlations between areas were retinotopically organized. Our data provide the developmentally earliest account of the global functional organization of the visual system, and demonstrate substrates for the adult organization across the whole visual system, not just thalamo-receipient V1 as shown previously (*Coogan and Van Essen, 1996*; *Horton and Hocking, 1997*; *Burkhalter et al., 1993*; *Blasdel et al., 1995*).

Areal distinctions and retinotopy were present throughout visual cortex in our earliest data; 10 days of age. It is unlikely that this organization emerged postnatally as a result of visually driven activity. Given the rate of axonal growth (0.4–1.4 mm/day) (*Lallemend et al., 2012*), the wiring of visual cortex needed to support our results most certainly began prenatally. Our finding that arealization is functionally present at birth is consistent with previous anatomical studies in macaques showing that connections within occipital visual areas, V1 and V2, first develop around 100 days of gestation and that adult-like organization is evident later in gestation (*Coogan and Van Essen, 1996*; *Killackey and Chalupa, 1986*; *Barone et al., 1996*; *Batardière et al., 2002*) as well as within the first weeks of birth (*Baldwin et al., 2012*; *Horton and Hocking, 1996*). By measuring functional organization of the whole visual system at once, we show that the large-scale organization of the visual system throughout ventral temporal and dorsal parietal cortex is established prenatally, prior to visual experience. How these maps are established is beyond the scope of this study, but must involve a combination of molecular cues and activity-dependent sorting (*McLaughlin and O'Leary, 2005*; *Rosa, 2002*).

Functional organization was apparent at birth throughout the visual system, even though the visual system is still anatomically and functionally immature by several criteria. There is no consensus on what would define a 'mature' visual area—anatomy, physiology, connectivity, size, or malleability? In neonates, V1 exhibits a high degree of adult-like connectivity (*Coogan and Van Essen, 1996*; *Horton and Hocking, 1997*; *Burkhalter et al., 1993*), impressive adult-like modular organization (*Blasdel et al., 1995*) and precise adult-like selectivity to visual features (*Wiesel and Hubel, 1974*). Yet V1 is clearly immature at birth, given how dramatically modifiable young (but not adult) V1 is by postnatal visual experience (*Wiesel, 1982*). Therefore studies in infant extrastriate cortex showing adult-like physiological properties and connectivity (*Rodman et al., 1993a*, *1993b*) should not be interpreted as proving that these areas are mature and unmodifiable by experience. There is also evidence that both the physiology and anatomy of the extrastriate visual system is immature at birth (*Zhang et al., 2005*; *Baldwin et al., 2012*; *Zhang et al., 2013*; *Bourne and Rosa, 2006*; *Kiorpes and Movshon, 2003*, *Kiorpes and Movshon, 2004*, *Kiorpes and Movshon, 2014*). Further, the modular organization of IT can be altered by early experience (*Srihasam et al., 2014*; *Op de Beeck et al., 2008*; *Srihasam et al., 2012*). With so much evidence that extrastriate visual areas must be immature at birth, one might have expected not to find widespread functional organization within the first postnatal week. Yet we did find retinotopic organization and arealization as early as 10 days of age across the entire cortical visual hierarchy. Given the extensive anatomical and physiological changes that occur over the first postnatal year, the organization we describe must constitute a proto-architecture that subsequently undergoes maturation and refinement as a consequence of postnatal visual experience.

Neonatal monkeys exhibited a series of hierarchically organized retinotopic maps. The adult visual cortex contains a series of retinotopic maps; each map comprises a two dimensional gradient of visual field location. Correlated activity in neonates similarly varied as a function of visual field representation both along polar angle and eccentricity dimensions (*Figure 6 and 7*). In adults, each map in the hierarchy is not just a copy of the preceding map; rather both stimulus selectivity and receptive-field scale increase moving up the hierarchy (*Hubel and Wiesel, 1974*). This convergence with increasing receptive field scale is an architectural feature of the visual hierarchy in adults that is thought to be an important computational principle underlying both increasing stimulus selectivity and invariance to scale and position (*Riesenhuber and Poggio, 1999*). The hierarchical maps in infants similarly do not reflect a simple serial duplication from one map to the next, matched to scale, but the increasing average spatial frequency preference moving up the hierarchy indicates an increase in receptive field scale, as in adults. Thus our data indicate that the scaffolding for hierarchical computations thought to support object recognition is already present at, or emerges shortly after, birth.

Retinotopic correlation patterns were evident in parts of neonatal inferior temporal cortex that would eventually become selective for object categories, like faces or scenes. In adult humans and macaques, there is a correspondence between foveal bias and face selectivity and between peripheral bias and scene selectivity (*Lafer-Sousa and Conway, 2013*; *Hasson et al., 2002*). Recently, we found that category selectivity emerges in monkeys at around 6 months of age (~200 days) (*Livingstone et al., 2017*). Here, we show that the IT regions that eventually become face selective have a foveal bias at birth. Further, spatial frequency tuning, and by extension receptive field

properties that scale with eccentricity, was identified throughout visual cortex several months before face clusters emerged. This means that different parts of IT would have receptive fields that were already differentially selective for low-level stimulus features, before they became selective for higher-level features, such as category. Thus the correlation between eccentricity and receptive-field size, and between eccentricity and curvature means that a retinotopic proto-map is also a proto-map for low-level stimulus features. Faces differ from most other object categories in being rich in curved contours, whereas scenes are richer in rectilinearity (*Torralba and Oliva, 2003*), so a shape-biased proto-map could contribute to the stereotyped localization of face selectivity to foveal regions of IT (*Srihasam et al., 2014*). We tend to foveate faces but not scenes; so viewing behavior could also contribute to the stereotyped localization of category domains (*Hasson et al., 2002*; *Levy et al., 2001*).

Our data provide novel evidence supporting the hypothesis that a retinotopic proto-organization in IT serves as the scaffolding for subsequent category-selective organization, because it carries with it a shape-based organization and/or because of where in the visual field different categories are preferentially experienced. We propose that the retinotopic organization of visual cortex emerges very early in development as a consequence of molecular cues and self-organizing principles and serves as scaffolding for further experience-dependent specialization. At a broader scale, we propose that parallels exist for other sensory areas, and possibly all of cortex. For example, somatotopic organization was also apparent in the newborn data. Thus topographic maps in general may form the proto-architecture of cortex on which experience-dependent specializations are elaborated; this is likely to be a general principle in development.

## Materials and methods

### Monkeys

Functional MRI studies were carried out on 4 infant/juvenile Macaca mulattas, three female (B2, B3 and B4) and one male (B1), all born in our laboratory. Monkeys B1 and B2 correspond to B4 and B3, respectively, in *Livingstone et al. (2017)*. All procedures were approved by the Harvard Medical School Animal Care and Use Committee and conformed with National Institutes of Health guidelines for the humane care and use of laboratory animals (Protocol# IS00000888). For scanning, the monkeys were alert, and their heads were immobilized using a foam-padded helmet. We used helmets of increasing size as the monkeys grew. For the youngest monkeys we used a chinstrap with a rubber nipple to hold the jaw and deliver formula; when they were around 3 months of age we switched to a bite bar that delivered sweet juice. The monkeys were scanned in a primate chair that was modified to accommodate small monkeys in such a way that they were positioned upright when they were less than 2 months old, but positioned semi-upright, or in a sphinx position as they got larger. They were always positioned so that they could move their bodies and limbs freely, but their heads were restrained in a forward-looking position by the padded helmet. The monkeys were rewarded with formula or juice for looking at the screen. Gaze direction was monitored using an infrared eye tracker (ISCAN, Burlington, MA).

### Resting scans

During rest scans, the lights were turned off in the scanner and the monkeys slept or sat quietly and no visual stimuli were presented. Rest scans lasted between 4 min. 10 s and 6 min. A total of 9 (47 mins. 44 s) runs were collected in monkey B1 at 10 days old and 10 (55 mins. 10 s) runs were collected in monkey B2 at 18 days old. A total of 8 runs (40 mins.) of resting state scans were also collected in both monkeys B1 and B2 at 3 years of age for the eccentricity correlation analysis (see below).

### Stimuli

The visual stimuli were projected onto a screen at the end of the scanner bore.

#### Retintopic mapping

Retinotopic mapping was performed in the same monkeys at >1.5 years of age when they were able to maintain fixation for extended periods of time.

To obtain polar angle maps, visual stimuli consisted of a wedge that rotated either clockwise or counterclockwise around a central fixation point. The wedge spanned 0.5–10° in eccentricity with an arc length of 45° and moved at a rate of 9°/s. The wedge consisted of a colored checkerboard with each check's chromaticity and luminance alternating at the flicker frequency of 4 Hz. For details, see (*Arcaro et al., 2011*). Each run consisted of eight cycles of 40 s each. 10–12 runs were collected with an equal split in the direction of rotation.

To obtain eccentricity maps, visual stimuli consisted of an annulus that either expanded or contracted around a central fixation point. The duty cycle of the annulus was 10%; that is, any given point on the screen was within the annulus for only 10% of the time. The annulus swept through the visual field linearly. The ring consisted of the same colored checkerboard as the wedge stimulus. Each run consisted of seven cycles of 40 s each with 10 s of blank, black backgrounds in between. Blank periods were inserted to temporally separate responses to the foveal and peripheral stimuli. 10–12 runs were collected with an equal split in direction.

## Face vs object localizer: Mapping was performed in monkeys B1 and B2

Localizers were performed in both monkeys >1.5 years of age, after face patches in IT showed clear selectivity for faces vs objects (*Livingstone et al., 2017*). Each scan comprised blocks of images of either monkey faces or familiar objects on a pink noise background; each image subtended 20 × 20° of visual angle and was presented for 0.5 s; block length was 20 s, with 20 s of a neutral gray screen between image blocks. Blocks and images were presented in shuffled order. Images were equated for spatial frequency and luminance using the SHINE toolbox (*Willenbockel et al., 2010*). 5–26 runs were collected per scan session. Both gaze direction and the average V1 response were used to evaluate the data. For additional details, see *Livingstone et al. (2017)*.

## Spatial frequency mapping:

Spatial frequency mapping was performed in monkey B3 at 110 days of age. Visual stimuli consisted of 8 spatial frequency ranges: 3, 2, 1, 0.8, 0.5, 0.4, 0.2, 0.1 cycles per degree (cpd). Ten images were presented per category. Each image subtended 20 × 20° of visual angle and was presented for 0.5 s. Each cycle comprised a full sweep across spatial frequencies. Within a given run, spatial frequencies were presented in either ascending or descending order. Each run consisted of eight cycles of 40s each with 10s of blank, black background in between. Blank periods were inserted to temporally separate responses to the highest and lowest spatial frequencies. 12 runs were collected with an equal split between ascending and descending sweeps.

## Curved vs. rectilinear localizer

Mapping was performed in monkey B4. Straight patterns were chosen to represent a variety of rectilinear patterns. Curved patterns were generated by adding waves to the rectilinear patterns. Each image subtended 20 × 20° of visual angle and was presented for 0.5 s; block length was 20 s, with 20 s of a neutral gray screen between image blocks. Each condition was presented once per run. 30 runs were collected. For additional details, see *Srihasam et al. (2014)*.

### Scanning

Monkeys were scanned in a 3 T TimTrio scanner with an AC88 gradient insert using 4-channel surface coils (custom made by Azma Maryam at the Martinos Imaging Center). Each scan session consisted of 10 or more functional scans. We used a repetition time (TR) of 2 s, echo time (TE) of 13 ms, flip angle of 72°, ipat = 2, 1 mm isotropic voxels, matrix size 96 × 96 mm, 67 contiguous sagittal slices. To enhance contrast (*Leite et al., 2002*; *Vanduffel et al., 2001*) we injected 12 mg/kg monocrystalline iron oxide nanoparticles (Feraheme, AMAG Parmaceuticals, Cambridge, MA) in the saphenous vein just before scanning.

### General preprocessing

Functional scan data were analyzed using AFNI (https://afni.nimh.nih.gov/afni/; RRID:nif-0000-00259) (*Cox, 1996*), SUMA (https://afni.nimh.nih.gov/Suma) (*Saad and Reynolds, 2012*), Freesurfer (http://surfer.nmr.mgh.harvard.edu/; RRID:nif-0000-00304) (*Dale et al., 1999*; *Fischl et al., 1999*), JIP Image Analysis Toolkit (written by Joeseph Mandeville; http://www.nmr.mgh.harvard.edu/~jbm/jip/)

and Matlab (the Mathworks Inc., Natick MA; RRID:nlx_153890). Each scan session for each monkey was analyzed separately. All images from each scan session were aligned to a single timepoint for that session, detrended and motion corrected using AFNI. Data were spatially filtered using a Gaussian filter of 2 mm full-width at half-maximum (FWHM) to increase the signal-to-noise ratio (SNR) while preserving spatial specificity. Each scan was normalized to its mean. Data were registered using a two-step linear then non-linear alignment approach (JIP toolkit) to a high-resolution (0.5 mm) anatomical image acquired in each monkey (>1.5 years). First, neonate monkey brains were scaled by 130% in XYZ directions to match their 2-3 year old anatomical images and a 12-parameter linear registration was performed between the mean EPI image for a given session and a high-resolution anatomical image of that monkey when the monkey was 2–3 years of age. Next, a nonlinear, diffeomorphic registration was conducted. After registration, regions of Interest (ROIs) were projected from high-resolution anatomical volumes to in-session EPI images. ROIs were then manually edited to ensure conformity to the grey matter for in-session EPIs. Despite the relatively smaller size of the newborn brain, sampling area of individual voxels remained at a high enough resolution to differentiate individual areas for subsequent analyses.

## Retinotopy analysis

A Fourier analysis was used to identify spatially selective voxels from the polar angle and eccentricity stimuli (*Engel et al., 1997*; *Bandettini et al., 1993*). For each voxel, the amplitude and phase – the temporal delay relative to the stimulus onset – of the harmonic at the stimulus frequency were determined by a Fourier transform of the mean time series. To correctly match the phase delay of the time series of each voxel to the phase of the wedge or ring stimuli, and thereby localize the region of the visual field to which the underlying neurons responded best, the response phases were corrected for a hemodynamic lag (4 s). The counterclockwise (expanding) runs were then reversed to match the clockwise (contracting) runs and averaged together for each voxel. An F-ratio was calculated by comparing the power of the complex signal at the stimulus frequency to the power of the noise (the power of the complex signal at all other frequencies). Data were threshold at $p < 0.001$. 19 visual field maps (*Figure 1—figure supplement 1*) were identified in accordance with previous studies using comparable methods (*Janssens et al., 2014*; *Arcaro et al., 2011*; *Kolster et al., 2009*): V1, V2, V3, V4, V4A, MT, MST, FST, V4t, OTd, PITd, PITv, V3A, DP, CIP1, CIP2, LIP. Our definition of area PITd extended from its posterior border with OTd anterior and ventral to the anterior border of PITv. We recently identified two additional visual field maps, OTS1 and OTS2, in the occipitotemporal sulcus (*Arcaro and Livingstone, 2017*). Areas CIP1 and CIP2 were grouped together, as were areas V3A and DP and areas OTS1 and OTS2, yielding 16 seed areas for all subsequent correlation analyses.

## Correlation analysis

For correlation analyses, several additional steps were performed on the data: (1) removal of signal deviation >2.5 SDs from the mean (AFNI's 3dDespike); temporal filtering retaining frequencies in the 0.01–0.1 Hz band; (3) linear and quadratic detrending; and (4) removal by linear regression of several sources of variance: (i) six motion parameter estimates (three translation and three rotation) and their temporal derivatives, (ii) the signal from a ventricular region, (iii) the signal from a white matter region. Ventricular and white matter regions were identified on the mean EPI image for each scan session. Global mean signal removal was not included in the analysis. Each scan was normalized to its mean. To minimize the effect of any evoked response due to the scanner onset, the initial 5 TRs were removed from each rest scan. Regions of interested for correlation analyses were defined based on retinotopic mapping.

### Temporal correlation analyses

Pearson correlation analyses were performed within and across hemispheres on the 16 retinotopic seed areas defined above. The normalized mean signal for each visual field map was calculated for each scan and concatenated. To identify within-hemisphere and across-hemisphere area x area correlations, the mean activity of each cortical area was correlated with the mean activity of every other retinotopic area in the ipsilateral and contralateral hemispheres, respectively. *Figure 2a* shows the across-hemisphere area x area correlation matrix. Across-hemisphere correlations were also

calculated between the mean activity of dorsal and ventral quadrants for seed areas V1, V2, V3, V4, and V4A (*Figure 6a*). To identify across-hemisphere area x voxel correlations, the mean activity of each retinotopic area was correlated with the activity of each voxel in the contralateral hemisphere. To balance the number of datapoints (voxels) included in deriving the mean signal for each area, all areas were subsampled to the area with the smallest size (number of voxels), and the mean activity was then calculated and used for correlations. 100 iterations of this subsampling were performed and the average correlation coefficient across iterations was computed.

Areal correlation profile analysis

We then applied a second Pearson correlation analysis, referred to as the areal correlation profile, (*Figure 3*) to identify cortical areas whose profile of within-hemisphere correlations with all other cortical areas was similar to the across-hemisphere correlation profiles of individual voxels in the contralateral hemisphere. To do this, we correlated the within-hemisphere area x area correlation matrix with the across-hemisphere area x voxel correlation matrix. Effectively, for each voxel, the profile of correlations across all contralateral retinotopic areas was correlated with the profiles of each area's within-hemisphere area x area correlations from the hemisphere contralateral to each voxel. The within-hemisphere area x area correlations served as a template of the connectivity profile for each cortical area. To indicate, for a given area's profile, which area should have the strongest correlation, area self-correlations (having a value of 1) were included. When applying the template to across-hemisphere correlations, this tests the idea that homotopic areas will have the strongest correlations. This areal correlation profile approach yielded a voxel-wise measurement (map) of the similarity between each retintopic area's within-hemisphere correlation profile and every voxel's between-hemisphere correlation profile. The original temporal correlations at r > 0.15 (t > 5.5; p<0.0001, FDR-corrected) were used as the threshold to ensure that a given voxel's activity was similar to that of the seed area (*Figure 4*; *Figure 4—figure supplement 1*). See (*Arcaro et al., 2015*) for similar approach.

## Simulation of correlation due to instrumentation and analyses

Local signals between nearby voxels (<3–4 mm) within hemisphere contain some degree of correlation simply due to the intrinsic spatial spread of the fMRI signal (*Smirnakis et al., 2007*). To avoid such biases, we focused our analyses on the organization of across-hemisphere correlations, leveraging the fact that contralateral anatomical connections exist between homologous regions in adult monkey and human visual cortex (*Dehay et al., 1986*) and that across-hemisphere connectivity has proven valid in reflecting the known organization of adult visual cortex (*Van Essen et al., 1982*; *Kennedy and Dehay, 1988*). Although it should be apparent that the intrinsic spatial spread of the fMRI signal would not interfere with across-hemisphere correlations, we conducted a simulation analysis where we generated random noise datasets matching the temporal length and spatial volumes of the real data acquired in each animal, spatially smoothed the noise data to approximately match the intrinsic spread of the fMRI signal, 3.5 mm, then passed these data through the same pre-processing and correlation analysis pipeline as the real data. As expected, there were no significant correlations in this simulation between visual areas across hemispheres. For within-hemisphere correlations, there were significant correlations between nearby visual areas, though the r coefficients were on average 64% that of the real data for adjacent areas and 20% for nonadjacent areas that border the same area (e.g., between V1 and V3). Within-hemisphere correlations between distal areas were <10% that of real data and non-significant.

## Community structure and clustering

To assess the structure of areal correlations, the high-dimensional 16 × 16 areal seed correlation matrix (*Figure 2a*; averaged across monkeys and hemispheres) was converted to a dissimilarity (Euclidean distance) matrix and non-classical multi-dimensional scaling was applied with Kruskal's normalized criterion (MATLAB's mdscale). The first two principal dimensions were visualized (*Figure 5*). Clustering of the data was performed using (1) a spectral (eigendecompositon) algorithm (*Newman, 2006*) from the Brain Connectivity Toolbox (*Rubinov and Sporns, 2010*) and (2) grouping the data into an agglomerative hierarchical tree algorithm and clustering based on (Euclidean) distances. The spectral community detection algorithm automatically subdivides a (weighted) network

into non-overlapping groups that maximize the number of within-group links and minimize the number of between group links. This approach yields a metric (Q coefficient) of the degree of modularity of the functional network. The hierarchical cluster tree was created using Ward's linkage. In contrast to the spectral clustering, the hierarchical tree clustering approach requires a cutoff to be set. This cutoff is based on an inconsistency metric that characterizes each link of the tree by comparing its height with the average height of other links at the same level of the hierarchy (i.e., higher values mean less similar). For illustrative purposes, the cutoff was set to a value of 1, which yielded five clusters. This cutoff corresponded to the lowest cutoff before individual clusters comprised single areas. Clustering from higher and lower cutoffs can be inferred by visual inspection of the hierarchical tree. For the hierarchical tree clustering approach, data were clustered directly on the dissimilarity matrix and were visualized as a dendrogram that illustrates the functional distances between areas. This approach yields a metric (cophenetic coefficient) that represents how well the hierarchical tree structure captures the distances between areas in a network. For both approaches, analyses were performed on averaged monkey data, though results were similar in each individual monkey.

### V1 eccentricity bin analysis

For each hemisphere, V1 was divided into 8 bins evenly spaced in cortical distance along the eccentricity axis up to 10 degrees. Correlations were computed between each bin and all voxels in the ipsilateral hemisphere. To balance the number of datapoints (voxels) included in deriving the mean signal, bins were subsampled to the lowest bin size (number of voxels), and the mean time-series was calculated, and then correlated with the time-series of every voxel. 100 iterations of this subsampling were performed and the average correlation coefficient across iterations was used for each bin to derive an estimate of the correlation between each bin and the rest of the brain. Voxels whose variance was significantly explained by some combination of eccentricity bins were included in subsequent analyses (assessed by linear regression; $F > 20$; $p<0.0001$). The preferred eccentricity bin for each voxel was assessed by fitting a modified Gaussian function:

$$A0 + A1 \, {}^* \exp(-((x - B1)/C1)^{\wedge}2),$$

where A0 equals a y-axis offset, A1, corresponds to the height of the curve from trough to peak, B1 equals the x-axis offset (i.e., the preferred eccentricity bin), and C1 is the width of the curve. Data were further threshold such that voxels had a minimum correlation coefficient (r) of 0.15 ($t > 5.5$; $p<0.0001$) at the bin closest to the peak and were color-coded based on the peak of the fit.

### Face vs. object analysis

A multiple regression analysis (*Cox, 1996*) in the framework of a general linear model (*Friston et al., 1995*) was performed on the experimental data. Each stimulus condition was modeled with a MION-based hemodynamic response function. Additional regressors that accounted for variance due to baseline shifts between time series, linear drifts, and head motion parameter estimates were also included in the regression model. Due to the time-course normalization, resulting beta coefficients were scaled to reflect % signal change. Face-minus-object contrast maps were threshold at $p<0.05$ FDR corrected. Average maps for four scan sessions were derived by thresholding and binarizing each session, then computing a voxelwise overlap of significant voxels across sessions.

### Spatial frequency analysis

A Fourier analysis was used to identify voxels selective for particular spatial frequencies. Analysis pipeline mirrored that described in Materials and methods: Retinotopy Analysis. Data were threshold at $p<0.001$. Data were binned by visual area and averaged to derive mean spatial frequency tunings. To evaluate spatial frequency as a function of eccentricity, data were grouped into 1° eccentricity bins within the central 8°. Due to the uneven distribution of spatial frequencies across eccentricity values from cortical magnification, linear fits were computed across the mean preferred spatial frequencies of each eccentricity bin.

### Curved vs. rectilinear analysis

A multiple regression analysis was performed on the experimental data as described in Materials and methods: Face vs. Object Analysis. Curved-minus-rectilinear contrast maps were threshold at

p<0.05 FDR corrected. To evaluate curvature tuning as a function of eccentricity, data were grouped into 1° eccentricity bins within the central 8° and linear fits were computed across the mean response contrast of curved-minus-rectilinear of each eccentricity bin.

## Acknowledgements

This work was supported by NIH grants RO1 EY 25670, P30 EY 12196, and F32 EY 24187. This research was carried out in part at the Athinoula A.Martinos Center for Biomedical Imaging at the Massachusetts General Hospital, using resources provided by the Center for Functional Neuroimaging Technologies, P41EB015896, a P41 Biotechnology Resource Grant supported by the National Institute of Biomedical Imaging and Bioengineering (NIBIB), National Institutes of Health, and NIH Shared Instrumentation Grant S10RR021110. We thank A Schapiro for helpful comments on the manuscript. We thank P Schade for monkey training and J Vincent for help in scanning.

## Additional information

### Funding

| Funder | Grant reference number | Author |
| --- | --- | --- |
| William Randolph Hearst Fellowship | | Michael J Arcaro |
| National Eye Institute | P30 EY 12196 | Margaret S Livingstone |
| National Institute of Biomedical Imaging and Bioengineering | P41EB015896 | Margaret S Livingstone |
| National Eye Institute | R01 EY 25670 | Margaret S Livingstone |

The funders had no role in study design, data collection and interpretation, or the decision to submit the work for publication.

### Author contributions

MJA, Conceptualization, Data curation, Software, Formal analysis, Validation, Investigation, Visualization, Methodology, Writing—original draft, Writing—review and editing; MSL, Conceptualization, Resources, Supervision, Funding acquisition, Investigation, Writing—original draft, Writing—review and editing

### Author ORCIDs

Michael J Arcaro, http://orcid.org/0000-0002-4612-9921

### Ethics

Animal experimentation: All procedures were approved by the Harvard Medical School Animal Care and Use Committee and conformed with National Institutes of Health guidelines for the humane care and use of laboratory animals. (Protocol# IS00000888).

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
