## [Decision Letter]

Thank you for submitting your article "A Hierarchical, Retinotopic Proto-Organization of the Primate Visual System at Birth" for consideration by *eLife*. Your article has been reviewed by two peer reviewers, and the evaluation has been overseen by a Reviewing Editor and David Van Essen as the Senior Editor. The following individual involved in review of your submission has agreed to reveal his identity: Wim Vanduffel (Reviewer #2).

The reviewers have discussed the reviews with one another and the Reviewing Editor has drafted this decision to help you prepare a revised submission.

Summary:

The manuscript provides a compelling demonstration of retinotopic functional topographic organization of the visual system in the newborn macaque brain. It shows for the first time that this organization contains adult-like hierarchical structure and does not require visual experience. The reviewers were enthusiastic in supporting the work, emphasizing its high quality, novelty, thorough analysis of the data and excellent writing.

However, the reviewers raised a few issues that should be addressed before the paper is accepted for publication. While these issues are not major, addressing them will undoubtedly improve the manuscript and its accessibility.

Essential revisions:

In response to Reviewer 1:

1) Please clarify Figure 3 and add the actual r-values in Figure 2, Figure 4 and Figure 5.

2) Address the comment about the use of the term "Spatial pattern correlations".

3) Clarify the sentence in the subsection “Retinotopic organization in newborn” starting with "these correlations likely reflect…"

4) If possible, address the question whether the newborn structural connectivity revealed by the functional connectivity analysis is different when performed in adults.

In response to Reviewer 2:

1) Address the lack of visually-driven activity raised by this reviewer.

2) The reviewer questions the validity of scaling the newborn brain without applying non-rigid registration. Please address.

3) Please explain the use of the "novel nomenclature" (PITd/PITv/ OT/aPIT).

4) Please provide more details of the fMRI experiments, with the exact description of each fMRI run, numbers of images acquired. Were all data analyzed? If not what were the exclusion criteria, etc.

*Reviewer #1:*

Arcaro and Livingstone present here a project investigating the development of retinotopy longitudinally in two infant and adolescent macaques. The authors employ functional connectivity of voxel timeseries to examine the extent to which arealization and retinotopy has emerged in macaques ~10 days old. They demonstrate that interhemispheric functional connectivity is capable of distinguishing different visual areas, and that a voxel's connectivity profile to the contralateral hemisphere well-matches its position in its respective hemisphere. They furthermore demonstrate that the functional connectivity during infancy already contains the hierarchical structure of adulthood and that this retinotopy may serve as a scaffold on which later category-selective regions emerge.

Overall, the study is thorough and very well done. This project is novel in that it extends our understanding of visual cortex organization into a very early developmental timepoint, and across the entire visual cortex. It also makes the important observation that regions of cortex that will later exhibit category-selectivity have a functional connectivity bias to certain eccentricity bands that precedes their category-selectivity. Retinotopy has for some time been a proposed organizational principle of high-level visual cortex and this study nicely offers strong evidence in its favor. I recommend this study for publication.

I have only a few minor comments and questions for the authors to address.

Subsection “Arealization of newborn visual cortex”: what is the t value associated with r values?

For Figure 2, Figure 4 and Figure 5, please put actual r-values and not min max on the colorbars.

It took some effort to understand what was going into Figure 3. A schematic illustrating the procedure might help. Below I describe what I think is going on. If it's wrong then the text might need clarification:

You correlate a voxel with the mean signal from each area in the other hemisphere. This gives you a between-hemisphere connectivity profile (BHCP). Separately, for each area in the other hemisphere you derive the areal correlation profile (ACP), which is an area's correlation with each other area in the ipsilateral hemisphere. You then take the voxel's between-hemisphere connectivity profile and correlate it with the ACP of an area in the other hemisphere, and then you color it according to "how much its profile looked like that area's profile" and you do this in a voxel for each area giving you Figure 3, which is why a voxel can take on different values in each map depending on which area its BHCP is being correlated with. However, a voxel will have a BHCP vector of length N (N=number of areas), however an area with will have an ACP vector of N-1 because you can't correlate it with itself. Did you just exclude that point in the vector when correlating? Or have I misunderstood something?

The phrase "spatial pattern correlations" is misleading: the pattern of activity in V1, for example, is not being correlated with anything. A better term would be something along the lines of profile correlation, since you're correlating profiles of connectivity.

In the subsection “Retinotopic organization in newborn”: what does this mean "these correlations likely reflect the underlying topographic organization within each area, not solely mirror symmetrical point-to-point connections between hemispheres"? Different quadrants don't have point-to-point connections, right? Except for maybe stitching together of the vertical meridian they share, is that what you mean?

The analysis and result in Figure 6 is striking. However, I'm curious to know what is the outcome of functional connectivity analysis done during infancy when repeated in adulthood. Is the connectivity structure unchanged, or is there a refinement of the connectivity structure? Either way the result is will be informative. If the adult functional connectivity looks closer to the retinotopic mapping data, then that suggests that inherent connectivity is refined with development, which is interesting. However, it functional connectivity remains the same as it was in infancy, it suggests that the connectivity provides a scaffold on which later retinotopic maps develop.

Figure 8 is very nice! I'm satisfied with it as is. A future analysis might be to make a model relating cycles-per-degree (CPD) tuning in the infant data to the population receptive field (pRF) size of a voxel in the adult data. That will enable using independent data in infancy to determine if one can predict the pRF size in adulthood within each voxel. This could provide strong evidence supporting the hypothesis that CPD is related to pRF size.

While I recognize this is beyond the scope of the current paper, it would be interesting in the future to try and estimate the population receptive field of each voxel as a weighted sum of timecourses from voxels in V1 or prior visual areas. Using the known transformation of the visual field to V1, one then could use this weighting to derive the pRF of every voxel in subsequent areas and estimate the pRF size and eccentricity and then compare it to adult data. Might be a nice future direction.

*Reviewer #2:*

Based on monkey fMRI data, Arcaro and Livingstone showed that the newborn visual system contains a topographic organization reflecting that observed in adult macaques. This proto-organization emerges before the development of face selectivity and they propose that it provides a scaffold for the development of the fully-developed visual system, including category selective patches.

This is a very well written manuscript based on a data set that is exceedingly challenging to obtain. I truly commend the authors for this achievement. The analyses are state-of-the-art and well-done. I truly enjoyed reading this manuscript that conveys a very important result which is difficult/impossible to obtain without whole-brain imaging. This message should be of interest to a broad readership not restricted to vision scientists – as the authors may have discovered a general developmental principle. I only have a few comments that should be taken into account before I can formally recommend publication for *eLife*.

1) An important (though not critical) result is the lack of visually-driven activity in cortex in newborns while rest correlations between V1 and extrastriate cortex are already profoundly present. Visually-driven activity becomes strong at an age of >30 days. I've several questions with regard to this finding:

A) The two measures are fundamentally different (regression versus correlation), which required quite different pre-processing steps. Did the authors consider age-dependent differences in the hemodynamic response that may explain this result? This could affect the regression analysis but not the correlation analysis -and it could explain the difference in visual responses between day 10 and 30. This strange GLM finding also conflicts with studies showing that many neurons show already stimulus selectivity around the time of natural eye opening, although weaker than in adults.

B) A mutually non-exclusive explanation may be that the newborns simply closed their eyes inside the scanner (due to anxiety?) – as it must have been their first time in a rather obnoxious environment for them. Were eye-movement recordings analyzed in these very young infants?

2) The authors scaled the newborn brain by 130% relative to match it with the older brain and non-rigid registration was not applied, as far as I can see. How confident are the authors that this procedure is valid? Most results hinge on the back-mapping of maps acquired at an older age to the newborn cortex, so this is an important analytical step that requires validation. Why not adding non-rigid registration procedures?

3) I'm puzzled why the authors decided to use a novel nomenclature for areas that have been described previously. Apparently, previously described OTd (Janssens et al. and Kolster et al. 2014) corresponds to the new PITd (present manuscript) and previous PITd with aPIT (present manuscript). There is no obvious reason to confuse the readership. New areas PITd/PITv/ OT/aPIT is supposed to form a cluster with a shared foveal representation, but the same holds true for V4A, PITd, PITv, and OTd as described in Janssens et al. The data presented in the present manuscript are not detailed enough to defend a new nomenclature.

[Editors' note: further revisions were requested prior to acceptance, as described below.]

Thank you for resubmitting your work entitled "A Hierarchical, Retinotopic Proto-Organization of the Primate Visual System at Birth" for further consideration at *eLife*. Your revised article has been favorably evaluated by David Van Essen (Senior Editor), a Reviewing Editor, and two reviewers.

The manuscript has been improved but there is a remaining issue that needs to be addressed before acceptance, as outlined below:

1) Please address the request from reviewer 1 to provide functional connectivity analysis in the older monkeys to enable the comparison with baby monkeys. This request was supported by reviewer 2, who in response to this request commented as follows: "Although the interpretation of the current data set does not depend on the additional analysis, it will considerably strengthen the paper".

*Reviewer #1:*

In this revision, the authors addressed most of the comments raised in the original review. Overall, the paper is strong and of interest for the readership of *eLife*. However, there is one remaining major concern.

In the prior review, one of the main concerns read: "The analysis and result in Figure 6 is striking. However, I'm curious to know what is the outcome of functional connectivity analysis done during infancy when repeated in adulthood. Is the connectivity structure unchanged, or is there a refinement of the connectivity structure? Either way the result is will be informative. If the adult functional connectivity looks closer to the retinotopic mapping data, then that suggests that inherent connectivity is refined with development, which is interesting. However, if functional connectivity remains the same as it was in infancy, it suggests that the connectivity provides a scaffold on which later retinotopic maps develop".

The authors replied "We are presently working on tracking the retinotopic organization of older/juvenile (2-3 years of age) monkeys. So far, the organization appears to be similar, though it is difficult to make direct comparisons with the correlation approach. To match these early neonate data to the 2-3 year old data, we needed to scale the brains by ~130%, which means our effective sampling resolution was coarser for the neonate data."

While the authors did not address this concern, we believe it is important to show the functional connectivity analyses in the older monkeys (>1.5 years) and compare it to the baby monkeys because the neonate monkeys cannot fixate and therefore all the analyses in the neonates are done with functional connectivity rather than retinotopy. As such, the authors compare one map (functional connectivity in the baby monkeys) to another map of eccentricity from retinotopic mapping (in the >1.5 year old monkeys). Given that the authors transformed the older monkeys' ROIs to the baby monkey brains, it seems a straightforward analysis to do the same functional correlation analysis on the older monkeys (on which these ROIs were defined in the first place). That the brain changes size across areas is a potential concern for all analyses, not just this one. Therefore, their argument against doing this analysis undermines the other analyses they are performing. The reason that measuring the functional connectivity in the older monkeys (e.g. Figure 2, Figure 5) is important is that this analysis will enable estimating what aspects of functional connectivity stay the same with age and what components develop, as described in the initial comment. Thus, the outcome of this analysis will flesh out what the authors mean by proto-retinotopic organization.

*Reviewer #2:*

The authors addressed all my concerns. This is a very neat and important paper, which will be of interest to many. I would like to commend the authors for addressing this challenging question!

[Editors' note: further revisions were requested prior to acceptance, as described below.]

Thank you for resubmitting your work entitled "A Hierarchical, Retinotopic Proto-Organization of the Primate Visual System at Birth" for further consideration at *eLife*. Your revised article has been favorably evaluated by David Van Essen (Senior Editor), a Reviewing Editor, and two reviewers.

The manuscript has been improved but there is one remaining issue that need to be addressed before acceptance, as outlined below:

Reviewer 1 points out that "Figure 6 only shows the resting state correlations in the neonates". Please address the comment that the updated Figure 6 should include the correlations in the juveniles.

*Reviewer #1:*

The authors had addressed my major remaining comment asking whether it is retinotopy that is developing, or that the relationship between retinotopy and resting state correlation that is developing. To address this concern I suggested comparing the resting state correlations in the juveniles compared to neonates.

They did a slightly different analysis than I suggested, which is fine with me. In their revision, they compared the correlations between retinotopic correlations and resting state functional correlations within the juveniles to the retinotopic correlations in juveniles vs. resting state correlations in the neonates.

They report the results in the subsection “Retinotopic organization in newborn”:

“Excluding V1, the mean absolute deviation between eccentricity correlations at newborn and juvenile ages was 2.0° across retinotopic areas, in both hemispheres, in both monkeys. Juvenile eccentricity correlations were more similar to the eccentricity mapping (mean deviation = 1.4°) than to the neonate eccentricity correlations, potentially indicating refinement of retinotopic maps over development. However, these differences might reflect non-biological variance (e.g., the precision of anatomical registration and proximity of coil placement due to brain size differences across ages). These data indicate that extensive retinotopic organization across both early and higher visual cortex was already present within the first weeks of life.”

What is still missing is a figure illustrating these results. In the response letter they write: "We include these new data in a revised Figure 6." However, I did not see these new data in Figure 6. Figure 6 only shows the resting state correlations in the neonates. Please update Figure 6 to include the correlations in the juveniles.

*Reviewer #2:*

The authors addressed the remaining issues raised by the other reviewer. I've no further comments.

---

## [Author Response]

*Essential revisions:*

*Reviewer #1:*

*[…] I have only a few minor comments and questions for the authors to address.*

*Subsection “Arealization of newborn visual cortex”: what is the t value associated with r values?*

We have made sure that all t values are now reported either in text or in figure legends. All correlation maps were threshold at r > 0.15, t > 5.66. The correspondence between r and t is also true for the correlation matrix in Figure 2 and the MDS plot in Figure 4. For Figure 2 and Figure 5 values are reported for homotopic correlations vs. zero as well as for comparisons between conditions.

*For Figure 2, Figure 4 and Figure 5, please put actual r-values and not min max on the colorbars.*

We have changed the min/max labels to the actual r values. We clarify in the legends that these correspond to the min and max correlations.

*It took some effort to understand what was going into Figure 3. A schematic illustrating the procedure might help. Below I describe what I think is going on. If it's wrong then the text might need clarification:*

We appreciate this feedback. It has been challenging to communicate this analysis effectively. We have a schematic illustrating the procedure in Figure 2—figure supplement 1. Note, this contains an illustration for both the areal pairwise correlations (referred to as Areal Correlation Profile) as well as the Across-hemisphere correlation profile.

*You correlate a voxel with the mean signal from each area in the other hemisphere. This gives you a between-hemisphere connectivity profile (BHCP). Separately, for each area in the other hemisphere you derive the areal correlation profile (ACP), which is an area's correlation with each other area in the ipsilateral hemisphere. You then take the voxel's between-hemisphere connectivity profile and correlate it with the ACP of an area in the other hemisphere, and then you color it according to "how much its profile looked like that area's profile" and you do this in a voxel for each area giving you Figure 3, which is why a voxel can take on different values in each map depending on which area its BHCP is being correlated with. However, a voxel will have a BHCP vector of length N (N=number of areas), however an area with will have an ACP vector of N-1 because you can't correlate it with itself. Did you just exclude that point in the vector when correlating? Or have I misunderstood something?*

Your summary is correct. In Figure 2—figure supplement 2, we refer to the “BHCP” as Across-hemisphere correlation profile. When comparing the BHCP with the ACP, we do use the full ACP vector (not N-1), even though the areal correlation with itself is 1. We consider the ACP a template of what the connectivity should be. By including the area self-correlation (having a value of 1) in the template, this serves the purpose of setting the correlation value greater than any other pairwise correlation and tests the idea that homotopic correlations will have the strongest correlations. While working on a previous experiment looking at correlations between visual cortex and the thalamus in humans (Arcaro et al. 2015 Journal of Neuroscience), the first author tried several variants of this approach: (1) using a spearman rank correlation, (2) calculating the self-correlation between sub-sampled, non-overlapping portions of a given area, (3) simply dropping the self-correlation (N-1) as the reviewer suggested. All alternative approaches yielded very similar results to the current approach used in this study. We have uploaded the correlation code to GitHub with a sample dataset and included options to compute with Spearman rank correlations and use an n-1 approach: https://github.com/mikearcaro/AreaProfileCorrelation.

*The phrase "spatial pattern correlations" is misleading: the pattern of activity in V1, for example, is not being correlated with anything. A better term would be something along the lines of profile correlation, since you're correlating profiles of connectivity.*

We thank the reviewer for this feedback. We actually had been using the term ‘profile correlation’ previously (Arcaro et al. 2015), but worried that would be too opaque to readers and wanted to emphasize that these correlations were not temporal, but rather across areas. We acknowledge that “spatial” may be misleading and now refer to these correlations as ‘profile correlations’. We have updated the text in the subsections “Arealization of newborn visual cortex” and “Correlation Analysis” accordingly.

*In the subsection “Retinotopic organization in newborn”: what does this mean "these correlations likely reflect the underlying topographic organization within each area, not solely mirror symmetrical point-to-point connections between hemispheres"? Different quadrants don't have point-to-point connections, right? Except for maybe stitching together of the vertical meridian they share, is that what you mean?*

That is correct; we were referring to the point-to-point correlations between overlapping receptive fields along the vertical meridian. Even in posterior occipital areas with relatively small RFs (e.g., V2 and V3) contralateral connections can spread up to several millimeters from the vertical meridian. To our knowledge the extent of the spread of such connections for higher order areas in temporal and parietal cortex have not been similarly quantified, but, given the relatively large receptive fields, we expect the spread (and link between homotopic quadrants) to be substantial. We now clarify this in the text with this additional comment:

“Even though right and left hemispheres represent left and right visual fields, respectively, colossal connections linking neurons with overlapping receptive fields at or near the vertical meridian (Dehay, Kennedy and Bullier, 1986) do generate correlated activity between hemispheres in adults(Heinzle, Kahnt and Haynes, 2011).”

*The analysis and result in Figure 6 is striking. However, I'm curious to know what is the outcome of functional connectivity analysis done during infancy when repeated in adulthood. Is the connectivity structure unchanged, or is there a refinement of the connectivity structure? Either way the result is will be informative. If the adult functional connectivity looks closer to the retinotopic mapping data, then that suggests that inherent connectivity is refined with development, which is interesting. However, it functional connectivity remains the same as it was in infancy, it suggests that the connectivity provides a scaffold on which later retinotopic maps develop.*

We are presently working on tracking the retinotopic organization of older/juvenile (2-3 years of age) monkeys. So far, the organization appears to be similar, though it is difficult to make direct comparisons with the correlation approach. To match these early neonate data to the 2-3 year old data, we needed to scale the brains by ~130%, which means our effective sampling resolution was coarser for the neonate data. While our registration was good, this inherently introduces extra smoothing. In addition, as the brain size grew, the distance of medial regions from the coil grew, effectively shifting the SNR profile across the brain. These factors can bias the data in varied ways and we feel that some correction would be needed to appropriately assess any differences. We agree that this would be a very interesting avenue to explore and feel that in addition relating cpd to pRF size (as the reviewer suggests next) will be a complementary approach.

*Figure 8 is very nice! I'm satisfied with it as is. A future analysis might be to make a model relating cycles-per-degree (CPD) tuning in the infant data to the population receptive field (pRF) size of a voxel in the adult data. That will enable using independent data in infancy to determine if one can predict the pRF size in adulthood within each voxel. This could provide strong evidence supporting the hypothesis that CPD is related to pRF size.*

We agree. It will be interesting to compute pRFs and compare to the CPD in both infant and adult monkeys. We feel this will likely be the most accurate approach for addressing the prior comment.

*While I recognize this is beyond the scope of the current paper, it would be interesting in the future to try and estimate the population receptive field of each voxel as a weighted sum of timecourses from voxels in V1 or prior visual areas. Using the known transformation of the visual field to V1, one then could use this weighting to derive the pRF of every voxel in subsequent areas and estimate the pRF size and eccentricity and then compare it to adult data. Might be a nice future direction.*

Definitely! In relation to the above comment, we plan to use an approach such as connective field modeling (e.g., Haak et al. 2013; Gravel et al. 2014; Bock et al. 2015) to compare the pRF estimates from the connective field modeling to direct pRF measurements in monkeys when they are older. As the reviewer said, this approach in conjunction with spatial frequency mapping could be a great way to track any changes to retinotopic organization across early development, though the challenges discussed above in making comparisons across different ages/brain sizes would also apply to this approach.

*Reviewer #2:*

*[…] 1) An important (though not critical) result is the lack of visually-driven activity in cortex in newborns while rest correlations between V1 and extrastriate cortex are already profoundly present. Visually-driven activity becomes strong at an age of >30 days. I've several questions with regard to this finding:*

*A) The two measures are fundamentally different (regression versus correlation), which required quite different pre-processing steps. Did the authors consider age-dependent differences in the hemodynamic response that may explain this result? This could affect the regression analysis but not the correlation analysis -and it could explain the difference in visual responses between day 10 and 30. This strange GLM finding also conflicts with studies showing that many neurons show already stimulus selectivity around the time of natural eye opening, although weaker than in adults.*

We did evaluate whether differences in the hemodynamic response could account for the apparent lack of visually-evoked activity in the early data. We conducted regression analyses with several shifts in the delay. No visually-evoked activity was observed when temporally shifting the MION function. We agree that this is in apparent conflict with previous electrophysiological data, but this may reflect differences between physiological measurements and metabolic measures: a similar lack of early activity has been observed using deoxyglucose (Distler et al. 1996), which is closer to the hemodynamic response, we measured. We do not think that the lack of early visually-evoked responses with imaging implies that there is a lack of neuronal activity. It is likely a combination of some physiological maturity and behavioral state (see below). We reported the lack of very early visual responsiveness in V1 in our recent paper (Livingstone et al. 2017: Development of the macaque face-patch system). To further address this point, we have added the following text:

“Though the lack of early visually-evoked fMRI activity is in apparent contrast with previous electrophysiological findings that demonstrated visual responsiveness in infants (Rust, Schultz and Movshon, 2002; Zhang et al., 2005; Endo et al., 2000), this may reflect differences between physiological and metabolic measurements as a similar lack of early activity has been observed using deoxyglucose (Distler et al., 1996).”

*B) A mutually non-exclusive explanation may be that the newborns simply closed their eyes inside the scanner (due to anxiety?) – as it must have been their first time in a rather obnoxious environment for them. Were eye-movement recordings analyzed in these very young infants?*

When measuring visual responses we used eye-tracking information and selected only blocks where the eyes were open. Achieving central fixation was not possible in these infant monkeys, so we settled for selecting blocks where the monkeys’ eyes were open and they were looking at the screen. Even in the “eyes-open” data (which is what was used for Figure 1), we did not see significant evoked responses in cortex (with the exception of peripheral V1), though we did find a very strong response in the LGN that were smaller or absent in the eyes closed runs. We have been conducting additional experiments to follow up on this, but we do not have anything more conclusive to report at this time. We think this is an important result to probe further, but not necessary for the current manuscript.

*2) The authors scaled the newborn brain by 130% relative to match it with the older brain and non-rigid registration was not applied, as far as I can see. How confident are the authors that this procedure is valid? Most results hinge on the back-mapping of maps acquired at an older age to the newborn cortex, so this is an important analytical step that requires validation. Why not adding non-rigid registration procedures?*

This is an important detail of our analysis pipeline and our methods needed clarification. We did use a non-rigid registration for all EPIs to anatomical volumes. We used Joseph Mandeville’s JIP toolkit. In our experience, JIP outperforms AFNI’s 3dQwarp and FSL’s FNIRT. To align a neonate’s EPI to an anatomical volume at an older age, we first scale the EPI brain by ~%130 in XYZ directions, we then perform a 12-parameter linear registration to get the volumes in rough correspondence. We then perform a nonlinear, diffeomorphic registration (all part of the JIP toolkit). To improve registration accuracy of ventral areas, we manually drew brain masks that removed the cerebellum for both EPIs and anatomicals prior to registration. Good back-projection of areal ROIs also requires good surface segmentation. Monkey surfaces were created using Freesurfer. After using their automated scripts for generating segmentations, surfaces were manually edited by the first author slice-by-slice in all three orientations to ensure proper segmentation of the grey and white matter. This is time consuming, but worth the investment. After projecting ROIs into the neonate EPI volumes, ROIs were manually edited to ensure conformity to the grey matter. We have expanded our description of the alignment in the subsection “General preprocessing”. We have also included a reviewer figure that illustrates the accuracy of registration.

This figure shows the alignment of EPI to the monkey’s own anatomy. Blue indicates surfaces and red white matter. Note the good registration of anatomical details such as the “bumps” in the STS along the A-P orientation) (sagittal slice), which can vary greatly across individuals.

*3) I'm puzzled why the authors decided to use a novel nomenclature for areas that have been described previously. Apparently, previously described OTd (Janssens et al. and Kolster et al. 2014) corresponds to the new PITd (present manuscript) and previous PITd with aPIT (present manuscript). There is no obvious reason to confuse the readership. New areas PITd/PITv/ OT/aPIT is supposed to form a cluster with a shared foveal representation, but the same holds true for V4A, PITd, PITv, and OTd as described in Janssens et al. The data presented in the present manuscript are not detailed enough to defend a new nomenclature.*

The reviewer has made a valid point. To maintain consistency with the prior literature, we have redefined the areas in accordance with Janssens and Kolster’s papers. We have re-run all affected analyses and updated all relevant figures.

The reviewer was correct in his interpretation of our areas. What we referred to as PITd is referred to as OTd in Janssens et al. and Kolster et al. (and OT is V4A). We found the organization of V4A to be more complicated than previously described. In our mapping data, V4A consistently fell in-between the posterior fovea shared by V1-V4 and the anterior fovea shared by PIT and OTd. In addition, we found that the vertical meridians comprising the anterior borders of V4A to fork such that part of that border bends posterior and part bends anterior with off meridian representations in-between. Taken together, it was not entirely clear to us that V4A should be wholly attributed to the anterior cluster with OTd and PITd/v. We opted to split V4A along this meridian fork such that cortex from the anterior border of V4 up to the posterior fork was labeled as V4A and the cortex between the forks was labeled OT. We then labeled this anterior half OT, since it straddled occipital and temporal cortex. We labeled OTd (Kolster and Janssens) as PITd, since it is located immediately dorsal to PITv with the border between them straddling the crown of the lower bank of the STS. For the anterior-most area previously described as PITd by Kolster and Janssens (we refer to as aPIT), we found in most hemispheres that the polar angle map extended ventrally to PITv. To us, this made sense, as the PIT/OT cluster would then form an enclosed, circular cluster (similar to MT). As the reviewer appropriately pointed out, we have not substantiated our claim for diverging from Kolster and Janssens’ proposed organization. The particular parcellation of these areas is not critical for the current study. We have taken the advice of the reviewer and adjusted our area labels to match the previous literature.

[Editors' note: further revisions were requested prior to acceptance, as described below.]

*Reviewer #1:*

*In this revision, the authors addressed most of the comments raised in the original review. Overall, the paper is strong and of interest for the readership of eLife. However, there is one remaining major concern.*

*In the prior review, one of the main concerns read: "The analysis and result in Figure 6 is striking. However, I'm curious to know what is the outcome of functional connectivity analysis done during infancy when repeated in adulthood. Is the connectivity structure unchanged, or is there a refinement of the connectivity structure? Either way the result is will be informative. If the adult functional connectivity looks closer to the retinotopic mapping data, then that suggests that inherent connectivity is refined with development, which is interesting. However, if functional connectivity remains the same as it was in infancy, it suggests that the connectivity provides a scaffold on which later retinotopic maps develop".*

*The authors replied "We are presently working on tracking the retinotopic organization of older/juvenile (2-3 years of age) monkeys. So far, the organization appears to be similar, though it is difficult to make direct comparisons with the correlation approach. To match these early neonate data to the 2-3 year old data, we needed to scale the brains by ~130%, which means our effective sampling resolution was coarser for the neonate data."*

*While the authors did not address this concern, we believe it is important to show the functional connectivity analyses in the older monkeys (>1.5 years) and compare it to the baby monkeys because the neonate monkeys cannot fixate and therefore all the analyses in the neonates are done with functional connectivity rather than retinotopy. As such, the authors compare one map (functional connectivity in the baby monkeys) to another map of eccentricity from retinotopic mapping (in the >1.5 year old monkeys). Given that the authors transformed the older monkeys' ROIs to the baby monkey brains, it seems a straightforward analysis to do the same functional correlation analysis on the older monkeys (on which these ROIs were defined in the first place). That the brain changes size across areas is a potential concern for all analyses, not just this one. Therefore, their argument against doing this analysis undermines the other analyses they are performing. The reason that measuring the functional connectivity in the older monkeys (e.g. Figure 2, Figure 5) is important is that this analysis will enable estimating what aspects of functional connectivity stay the same with age and what components develop, as described in the initial comment. Thus, the outcome of this analysis will flesh out what the authors mean by proto-retinotopic organization.*

We apologize for misunderstanding the reviewer’s original comment. We thought the reviewer was asking us to perform quantitative comparisons on the degree of retinotopic organization between young and older monkey data, which we do not think this is appropriate to perform on the correlation data for the reasons outlined previously. Given the above comment, it is now clear the reviewer was requesting us to just show that the retinotopic organization is present in the correlation patterns of older monkeys. This is a good suggestion. We agree that it is important to verify that the retinotopic correlation structure remains present in older monkeys.

We have collected comparable resting data in monkeys B1 and B2 (8 runs @ 40 mins. total) and have re-run the eccentricity correlation analysis. These monkeys are now “juvenile” (~3 years of age). We include these new data in a revised Figure 6. In these new juvenile correlations, we find a similar organization to the neonate correlation data. The mean absolute deviation (across all visual areas tested) between the neonate correlation data and the juvenile correlation data is 2.0 degrees, comparable to the 2.2 degrees mean deviation between the neonate correlation data and the eccentricity mapping at > 1.5 years. Interestingly, the mean deviation between the juvenile correlation data and the eccentricity mapping was only 1.4 degrees. It is enticing to infer that the greater similarity between the juvenile correlations and eccentricity mapping indicates a refinement of retinotopic organization over the first 3 years of development. However, we are hesitant to make such inferences on such a modest difference given that the brain shape of the 3-year-old correlation data is more similar to the retinotopy mapping data and the neonate data had to be enlarged by ~130% in each dimension. Further, changes in the SNR profile and coil positioning due to brain size differences can also affect correlations. Regardless, these data demonstrate that the retinotopic correlation patterns of visual cortex are indeed present in juvenile monkeys. We include this new data and a discussion of the possible interpretations in the Results, subsection “Retinotopic organization in newborn”, third paragraph.

We would also like to point out that the ICA analyses reported in Figure 1—figure supplement 2 and Figure 8—figure supplement 2 further demonstrate the presence of retinotopic activity patterns in older monkeys. In these figures, we showed that foveal and peripheral components were identified in the neonate data as well as when the same monkeys were older (both prior to and after the emergence of face patches). In contrast, a face patch system ICA was only identified AFTER we could reliably detect task-evoked face selectivity in these monkeys, indicating that the retinotopic organization is present at birth, and likely serves as the scaffolding for subsequent development as it remains present through development. We have expanded discussion of these ICA results in the last paragraph of the subsection “Retinotopic organization in newborn”.

To be clear, even though the retinotopic organization is present at birth and persists over development, we think it is entirely possible that the maps are refined through experience. This is an important question, but is best suited to a dedicated project on the topic.

[Editors' note: further revisions were requested prior to acceptance, as described below.]

*Reviewer #1:*

*The authors had addressed my major remaining comment asking whether it is retinotopy that is developing, or that the relationship between retinotopy and resting state correlation that is developing. To address this concern I suggested comparing the resting state correlations in the juveniles compared to neonates.*

*They did a slightly different analysis than I suggested, which is fine with me. In their revision, they compared the correlations between retinotopic correlations and resting state functional correlations within the juveniles to the retinotopic correlations in juveniles vs. resting state correlations in the neonates.*

*They report the results in the subsection “Retinotopic organization in newborn”:*

*“Excluding V1, the mean absolute deviation between eccentricity correlations at newborn and juvenile ages was 2.0° across retinotopic areas, in both hemispheres, in both monkeys. Juvenile eccentricity correlations were more similar to the eccentricity mapping (mean deviation = 1.4°) than to the neonate eccentricity correlations, potentially indicating refinement of retinotopic maps over development. However, these differences might reflect non-biological variance (e.g., the precision of anatomical registration and proximity of coil placement due to brain size differences across ages). These data indicate that extensive retinotopic organization across both early and higher visual cortex was already present within the first weeks of life.”*

In the Results section (corresponding to revised Figure 7), we did compare the resting state correlations in juveniles to neonates. We stated: "Excluding V1, the mean absolute deviation between eccentricity correlations at newborn and juvenile ages was 2.0 degrees across retinotopic areas, in both hemispheres, in both monkeys." In the legend for Figure 7, we further provide a breakdown of this deviation for different regions of visual cortex: "(1.8, 2.5, 1.7, and 2.5 for occipital areas, MT cluster, inferior temporal, and dorsal/parietal areas, respectively)."

As we understand this comment, the reviewer had a different analysis in mind than what we provided, but what we have provided (i.e., updated Figure 7) was sufficient. If the above figure clarification doesn't fully address the remaining concern and if the different analysis the reviewer had in mind addresses something substantially different from what we have provided, we would be happy to include in a revision. In that case, we ask that the reviewer provide more specific details on exactly what analysis he/she is thinking of to make sure we fully address it in this next revision.

We have made one addition to the “Retinotopic organization in newborns”, Results section.

We originally only reported across-hemisphere correlations between dorsal and ventral quadrant pairs. Within-hemisphere correlations were not originally reported due to the likely influence of the intrinsic spatial spread of the fMRI signal. While this issue was discussed earlier in the Results, we realized that this was not thoroughly explained in this section. As such, the reasoning for using across-hemisphere correlations for these analyses may be lost on some readers. We now first report the within-hemisphere correlations between quadrants then explain that these correlations may be partly biased due to spread of the fMRI signal. We now use this to better motivate the use of across-hemisphere correlations. No changes were made to the following across-hemisphere correlations section. The beginning of this section now reads:

“An even finer level of differentiation within individual neonatal visual areas was apparent as a series of retinotopic representations that could be inferred from correlation patterns. […] To further rule out the influence of cortical proximity on retinotopic correlations, across-hemisphere correlations were assessed between posterior areas V1, V2, V3, V4, and V4A as a function of their dorsal and ventral quadrants.”

*What is still missing is a figure illustrating these results. In the response letter they write: "We include these new data in a revised Figure 6." However, I did not see these new data in Figure 6. Figure 6 only shows the resting state correlations in the neonates. Please update Figure 6 to include the correlations in the juveniles.*

To address the reviewer's previous request that we "provide functional connectivity analysis in the older monkeys to enable the comparison with baby monkeys," we included a new panel in Figure 7 in which we show functional connectivity analysis in the same monkeys at both neonate and older ages. In the response letter we referred to this figure as Figure 6 (that was the original figure number), but because we moved another figure from supplementary to the main text (in response to a request from reviewer #1), the manuscript figure is now Figure 7. It was only the response letter that referred to this Figure as #6, the manuscript correctly refers to this Figure as #7. We are sorry for this mix-up. This should address reviewer #1's remaining concern: "Figure 6 only shows the resting state correlations in the neonates." Figure 7, not Figure 6, shows the resting state correlations (as a function of eccentricity) for both neonate and juvenile ages.